# Hfq CLASH uncovers sRNA-target interaction networks linked to nutrient availability adaptation

Ira Alexandra Iosub[1], Robert Willem van Nues[2], Stuart William McKellar[1], Karen Jule Nieken[2], Marta Marchioretto[3], Brandon Sy[4], Jai Justin Tree[4], Gabriella Viero[3], Sander Granneman[1]*

[1]Centre for Synthetic and Systems Biology, University of Edinburgh, Edinburgh, United Kingdom; [2]Institute of Cell Biology, University of Edinburgh, Edinburgh, United Kingdom; [3]Institute of Biophysics, CNR Unit, Trento, Italy; [4]School of Biotechnology and Biomolecular Sciences, University of New South Wales, Sydney, Australia

**Abstract** By shaping gene expression profiles, small RNAs (sRNAs) enable bacteria to efficiently adapt to changes in their environment. To better understand how *Escherichia coli* acclimatizes to nutrient availability, we performed UV cross-linking, ligation and sequencing of hybrids (CLASH) to uncover Hfq-associated RNA-RNA interactions at specific growth stages. We demonstrate that Hfq CLASH robustly captures *bona fide* RNA-RNA interactions. We identified hundreds of novel sRNA base-pairing interactions, including many sRNA-sRNA interactions and involving 3'UTR-derived sRNAs. We rediscovered known and identified novel sRNA seed sequences. The sRNA-mRNA interactions identified by CLASH have strong base-pairing potential and are highly enriched for complementary sequence motifs, even those supported by only a few reads. Yet, steady state levels of most mRNA targets were not significantly affected upon over-expression of the sRNA regulator. Our results reinforce the idea that the reproducibility of the interaction, not base-pairing potential, is a stronger predictor for a regulatory outcome.

*For correspondence:
sgrannem@ed.ac.uk

Competing interests: The authors declare that no competing interests exist.

## Introduction

Microorganisms are renowned for their ability to adapt to environmental changes by rapidly rewiring their gene expression program. These responses are mediated through integrated transcriptional and post-transcriptional networks. Transcriptional control dictates which genes are expressed (*Balleza et al., 2009*; *Martínez-Antonio et al., 2008*) and is well-characterised in *Escherichia coli*. Post-transcriptional regulation is key for controlling adaptive responses. By using riboregulators and RNA-binding proteins (RBPs), cells can efficiently integrate multiple pathways and incorporate additional signals into regulatory circuits. *E. coli* employs many post-transcriptional regulators, including small regulatory RNAs (sRNAs (*Waters and Storz, 2009*)), *cis*-acting RNAs (*Kortmann and Narberhaus, 2012*), and RNA binding proteins (RBPs) (*Holmqvist and Vogel, 2018*). The sRNAs are the largest class of bacterial regulators, working in tandem with RBPs to regulate their RNA targets (*Storz et al., 2011*; *Waters and Storz, 2009*). By base-pairing with their targets, small RNAs can repress or stimulate translation and transcription elongation and control the stability of transcripts (*Sedlyarova et al., 2016*; *Updegrove et al., 2016*; *Vogel and Luisi, 2011*; *Waters and Storz, 2009*).

Base-pairing interactions are often mediated by RNA chaperones such as Hfq and ProQ, which help to anneal or stabilize the sRNA and sRNA-target duplex (*Melamed et al., 2020*; *Melamed et al., 2016*; *Smirnov et al., 2017*; *Smirnov et al., 2016*; *Updegrove et al., 2016*).

Although Hfq is most frequently mentioned in association with sRNA-mediated regulation, it can also control gene expression independently of sRNAs in response to environmental changes (*Salvail et al., 2013*; *Sonnleitner and Bläsi, 2014*). In *Pseudomonas aeruginosa*, Hfq directly binds to mRNAs to repress translation in response to changes in nutrient availability, which relies on a protein co-factor Crc that acts cooperatively with Hfq to inhibit translation (*Pei et al., 2019*; *Sonnleitner and Bläsi, 2014*).

During growth in rich media, *E. coli* are exposed to continuously changing conditions, such as fluctuations in nutrient availability, pH and osmolarity. Consequently, *E. coli* elicit complex responses that result in physiological and behavioural changes such as envelope composition remodelling, quorum sensing, nutrient scavenging, swarming and biofilm formation. Even subtle changes in the growth conditions can trigger rapid adaptive responses.

Accordingly, each stage of the growth curve is characterised by different physiological states driven by the activation of different transcriptional and post-transcriptional networks. Moreover, growth phase dependency of virulence and pathogenic behaviour has been demonstrated in both Gram-positive and Gram-negative bacteria. In some cases, a particular growth stage is non-permissive for the induction of virulence (*Mäder et al., 2016*; *El et al., 2018*). Although the exponential and stationary phases have been characterised in detail (*Navarro Llorens et al., 2010*; *Pletnev et al., 2015*), little is known about the transition between these two phases. During this transition, the cell population starts to scavenge alternative carbon sources, which requires rapid remodelling of their transcriptome (*Baev et al., 2006a*; *Baev et al., 2006b*; *Sezonov et al., 2007*).

To understand sRNA-mediated adaptive responses, detailed knowledge of the underlying post-transcriptional circuits is required. In *E. coli*, hundreds of sRNAs have been discovered, and only a small fraction of these have been characterised. A key step to unravel the roles of sRNAs in regulating adaptive responses is to identify their target mRNAs. To tackle this at genome-wide level, high-throughput methods have been developed to uncover sRNA base-pairing interactions (*Han et al., 2017*; *Hör et al., 2018*; *Hör and Vogel, 2017*; *Lalaouna et al., 2015a*; *Melamed et al., 2016*; *Waters et al., 2017*).

To unravel sRNA base-pairing interactions taking place during the entry into stationary phase, we applied UV cross-linking, ligation and sequencing of hybrids (CLASH) (*Helwak et al., 2013*; *Kudla et al., 2011*) to *E. coli*. Firstly, we demonstrate that the highly stringent purification steps make CLASH a robust method for direct mapping of Hfq-mediated sRNA base-pairing interactions in *E. coli*. This enabled us to significantly expand on the sRNA base-pairing interactions found by RNase E CLASH (*Waters et al., 2017*) and RIL-seq (*Melamed et al., 2016*). Additionally, we identified a plethora of sRNA-sRNA interactions and potentially novel 3'UTR-derived sRNAs, confirming that this class of sRNAs is highly prevalent (*Chao et al., 2012*; *Chao et al., 2017*; *Chao and Vogel, 2016*; *Miyakoshi et al., 2015a*). The sRNA-mRNA interactions identified by CLASH have a strong base-pairing potential and are highly enriched for complementary sequence motifs, even those supported by only a few chimeric reads. We rediscovered known and identified novel sRNA seed sequences in the CLASH data, implying they represent genuine in vivo interactions. However, in many cases, over-expression of the sRNA did not significantly impact the steady state levels of putative mRNA targets. Although base-pairing potential is important, our results reinforce the notion that reproducibly detected interactions, are more likely to impact target steady-state levels (*Faigenbaum-Romm et al., 2020*).

## Results

### Hfq CLASH in *E. coli*

To unravel the post-transcriptional networks that underlie the transition between exponential and stationary growth phases in *E. coli*, we performed CLASH (*Helwak et al., 2013*; *Kudla et al., 2011*) using Hfq as bait (*Figure 1A*). To generate high-quality Hfq CLASH data, we made a number of improvements to the original protocol used for RNase E CLASH (*Waters et al., 2017*). Our Hfq CLASH protocol has several advantages over the related RIL-seq method (see Materials and methods and Discussion). As negative controls, replicate CLASH experiments were performed on the untagged parental strain. When combined, the control samples had ~10 times less single-mapping reads and contained only 297 unique chimeric reads, compared to the over 50,000 chimeras

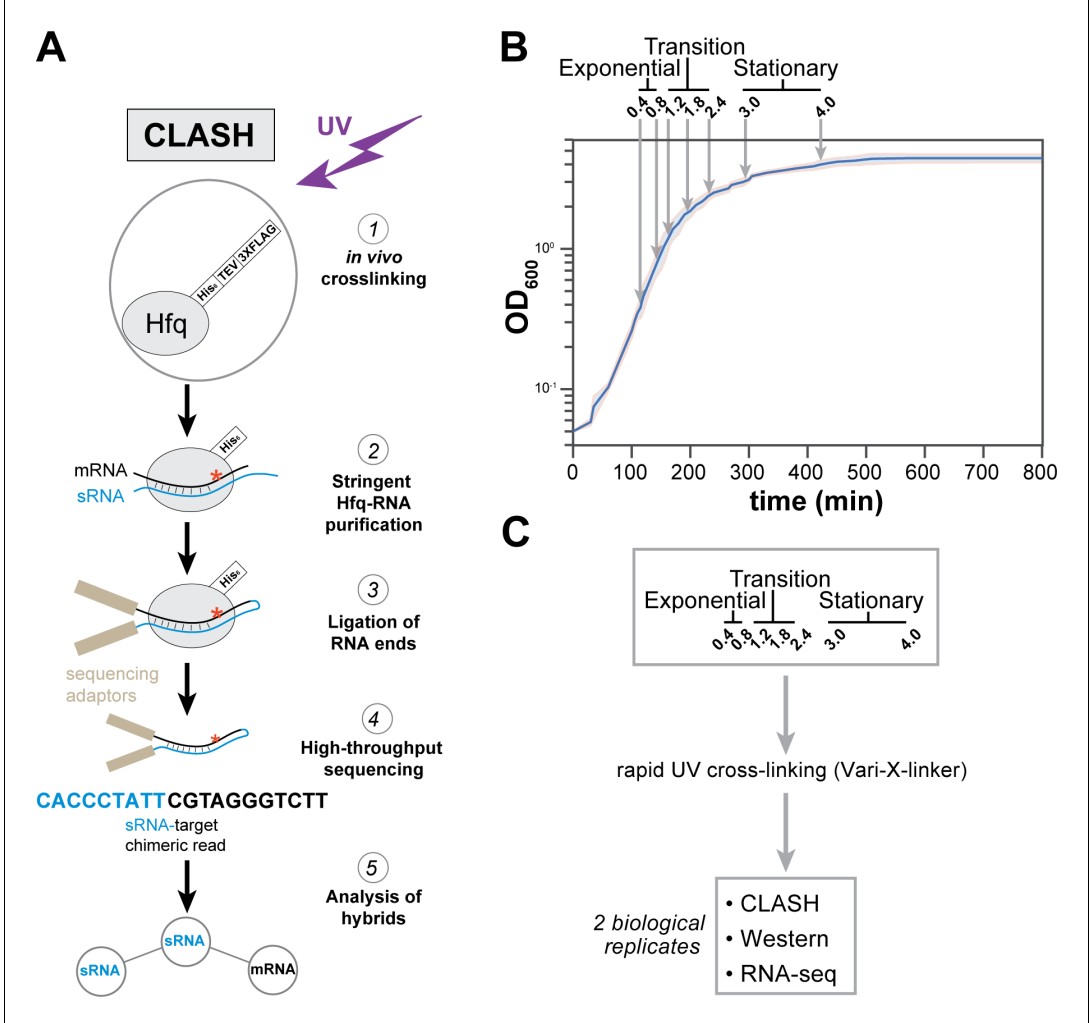

**Figure 1.** Hfq CLASH experiments at different growth phases in *E. coli*. (A) Overview of the critical experimental steps for obtaining the Hfq CLASH data. *E. coli* cells expressing an HTF (His6-TEV-3xFLAG)-tagged Hfq (*Tree et al., 2014*) were grown in LB and an equal number of cells were harvested at different optical densities (OD$_{600}$) and UV cross-linked. Hfq, cross-linked to sRNA-RNA duplexes is purified under stringent and denaturing conditions and RNA ends that are in close proximity are ligated together. After removal of the protein with Proteinase K, cDNA libraries are prepared and sequenced. The single reads can be used to map Hfq-RNA interactions, whereas the chimeric reads can be traced to sRNA-target interactions. (B) A growth curve of the cultures used for the Hfq CLASH experiments, with OD$_{600}$ at which cells were cross-linked. Each growth stage is indicated above the plot. The results show the mean and standard deviations of two biological replicates. Source data are provided as a Source Data file. (C) Cultures at the indicated OD$_{600}$ were cross-linked,harvested by filtration and analysed by Hfq CLASH, RNA-seq and western blotting to detect Hfq. All the experiments were done in duplicate.

The online version of this article includes the following source data and figure supplement(s) for figure 1:

**Source data 1.** Source data for *Figure 1B*.

**Figure supplement 1.** Hfq expression and Hfq binding to RNAs at different cell densities in UV-irradiated *E. coli*.

**Figure supplement 1—source data 1.** Source data for *Figure 1—figure supplement 1A and B*.

**Figure supplement 2.** RNA-seq and Hfq CLASH replicate datasets are highly correlated.

**Figure supplement 3.** Transcriptome-wide maps of Hfq binding to mRNA genes.

identified in the tagged Hfq data. This result demonstrates that the CLASH purification method produced very low background levels.

Cell samples from seven different optical densities were subjected to Hfq CLASH. Based on the growth curve analysis shown in *Figure 1B*, we categorized OD$_{600}$ densities 0.4 and 0.8 as exponential growth phase, 1.2, 1.8, 2.4 as the transition phase from exponential to stationary, and 3.0 and 4.0 as early stationary phase. To complement the CLASH data, RNA-seq and western blot analyses

were performed on UV-irradiated cells to quantify steady state RNA and Hfq protein levels, respectively (*Figure 1C*, *Figure 1—figure supplement 1*). Western blot analyses revealed that Hfq levels were very modestly increased during growth (*Figure 1—figure supplement 1A–B*). To determine the cross-linking efficiency, Hfq-RNA complexes immobilised on nickel beads were radiolabelled, resolved on NuPAGE gels and analysed by autoradiography. The data show that the recovery of Hfq and radioactive signal was comparable at each optical density studied (*Figure 1—figure supplement 1C*). Comparison of normalised read counts of replicate CLASH and RNA-seq experiments showed that the results were highly reproducible (*Figure 1—figure supplement 2*). Meta-analyses of the Hfq CLASH sequencing data revealed that the distribution of Hfq binding across mRNAs was very similar at each growth stage. We observed the expected Hfq enrichment at the 5'UTRs and at the 3'UTRs at each growth stage (*Figure 1—figure supplement 3A and B* for examples). After identifying significantly enriched Hfq-binding peaks (FDR <= 0.05; see Materials and methods for details), we used the genomic coordinates of these peaks to search for Hfq binding motifs in mRNAs. The most enriched k-mer included poly-U stretches (*Figure 1—figure supplement 3C*) that resemble the poly-U tracts characteristic to Rho-independent terminators found at the end of many bacterial transcripts (*Wilson and Von, 1995*), and confirms the motif uncovered in CLIP-seq studies in *Salmonella* (*Holmqvist et al., 2016*).

## Hfq CLASH robustly detects RNA-RNA interactions

To get the complete catalogue of the RNA-RNA interactions captured by Hfq CLASH, we merged the data from the two biological replicates of CLASH growth phase experiments (*Supplementary file 1*). Overlapping paired-end reads were merged and unique chimeric reads were identified using the hyb pipeline (*Travis et al., 2014*). To select RNA-RNA interactions for further studies, we applied a probabilistic analysis pipeline previously used for detecting RNA-RNA interactions in human cells (*Sharma et al., 2016*) and adapted for the analyses of RNase E CLASH data (*Waters et al., 2017*). This pipeline tests the likelihood that observed chimeras could have formed spuriously. Strikingly, 87% of the chimeric reads had a Benjamini-Hochberg adjusted p-value of 0.05 or less, indicating that it is highly unlikely that these chimeras were generated by random ligation of RNA molecules. A complete overview of statistically significantly enriched chimeras is provided in *Supplementary file 2*.

We next analysed the distribution of combinations of transcript classes found in the statistically filtered chimeric reads. Hfq CLASH identified over unique 2000 sRNA-mRNA target interactions represented by 18783 chimeras (*Figure 2A*; *Supplementary file 3*). These chimeras included sRNAs derived from 3'UTRs and were the most frequently recovered Hfq-mediated interaction type (65.7%; *Figure 2A*). We suspect that this number might be higher, as 1.7% of the chimeras contained fragments of sRNAs fused to short sequences from intergenic regions (*Figure 2A*). Manual inspection of several of these indicated that some of the intergenic sequences were located near genes for which the UTRs were either unannotated or too short. Interestingly, 10.5% of the intermolecular chimeras contained fragments from two different mRNAs (*Figure 2A*). Based on analyses presented below, we speculate that many of these could be interactions between novel 3'UTR-derived sRNAs and mRNA substrates. Around 1% of the chimeras represented sRNA-tRNA interactions. In *E. coli*, external transcribed spacers of tRNAs can base-pair with sRNAs to absorb transcriptional noise (*Lalaouna et al., 2015a*). In many cases, the predicted base-pairing interactions between the tRNA and sRNA halves in chimeras are quite extensive (*Supplementary file 2*). Hence, it is possible that this group contains biologically relevant interactions.

Most of the interactions, including sRNA-mRNA interactions, were identified in the transition phase (*Figure 2C–D*). The mRNA fragments found in chimeric reads were strongly enriched in 5'UTRs peaking near the translational start codon (*Figure 2E–F*), consistent with the canonical mode of translational inhibition by sRNAs (*Bouvier et al., 2008*). Enrichment was also found in 3'UTRs of mRNAs, although to a lesser extent compared to 5'UTRs (*Figure 2E*). Motif analyses revealed a distinct sequence preference in 5'UTR and 3'UTR binding sites (*Figure 2G–H*, *Supplementary files 8–9*). The motifs enriched in the 5'UTR chimeric fragments are more consistent with Hfq binding to Shine Dalgarno-like $(ARN)_n$ sequences (*Tree et al., 2014*; *Supplementary file 8*) and U-tracts, whereas the 3'UTR-containing chimera consensus motif corresponds to poly-U transcription termination sites (*Figure 2G–H* and *Supplementary file 9*).

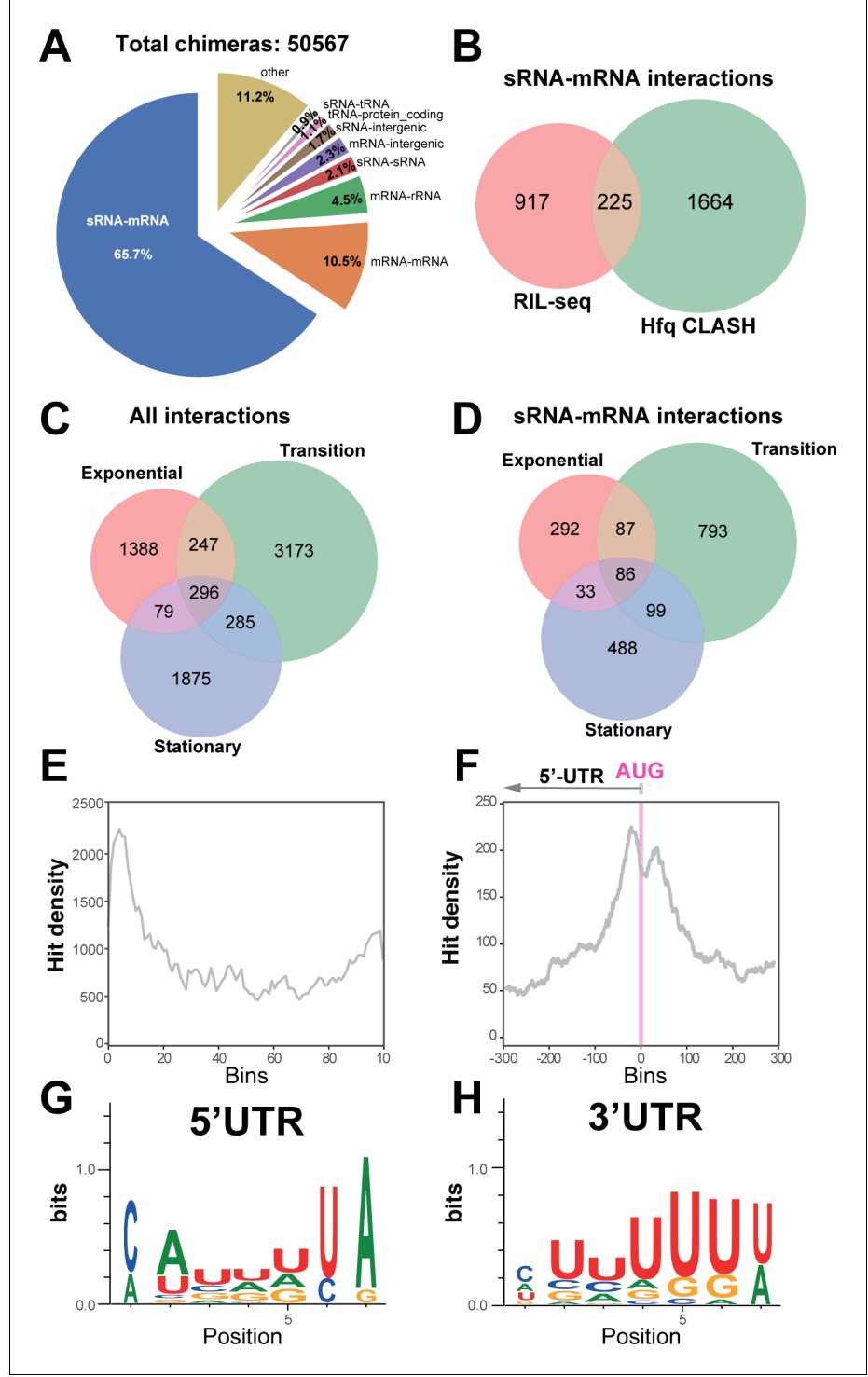

**Figure 2.** Hfq CLASH detects RNA-RNA interactions in *E. coli*. (**A**) Intermolecular RNA interactions found in chimeras captured by Hfq CLASH. Chimera counts for all the uniquely annotated hybrids that mapped to genomic features. *tRNA-tRNA and rRNA-rRNA chimeras originating from different genomic regions were removed because tRNA and rRNA gene copies are very similar and therefore we could not unambiguously determine if these represented intermolecular or intramolecular interactions. (**B**) Venn diagram comparing the sRNA-mRNA interactions found in RIL-seq S-chimera data (log and stationary) and Hfq CLASH data. (**C**) Venn diagram showing the intersection between interactions from statistically filtered CLASH data from two biological replicates, recovered at three main growth stages: exponential ($OD_{600}$ 0.4 and 0.8), transition ($OD_{600}$ 1.2, 1.8, 2.4) and early

*Figure 2 continued on next page*

*Figure 2 continued*

stationary (OD$_{600}$ 3.0 and 4.0). (**D**) Same as in (**C**) but for sRNA-mRNA interactions. (**E**) Distribution of mRNA fragments in sRNA-mRNA chimeras over all *E. coli* protein-coding genes. Each gene was divided in 100 bins and the number of mRNA fragments that mapped to each bin (hit density; y-axis) was calculated. (**F**) Distribution of the mRNA fragments of sRNA-mRNA chimeras around the translational start codon (AUG). The pink line indicates the position of the start codon (**G–H**) Enriched motifs in mRNA fragments of chimeras that uniquely overlap 5'UTRs and 3'UTRs; the logos were drawn using the top 20 K-mers.

The online version of this article includes the following figure supplement(s) for figure 2:

**Figure supplement 1.** Analysis of experimentally verified sRNA-mRNA chimeras in the Hfq CLASH data.
**Figure supplement 2.** sRNAs are most frequently found paired with mRNAs, and *vice versa,* in CLASH chimeras and are enriched for seed sequences.
**Figure supplement 3.** Interactions shared between RIL-seq and CLASH are supported by a large number of chimeras.
**Figure supplement 4.** sRNAs are most frequently found paired with mRNAs, and vice versa, in CLASH chimeras and are enriched in seed sequences.
**Figure supplement 5.** sRNAs are most frequently found paired with mRNAs, and vice versa, in CLASH chimeras and are enriched in seed sequences.

---

To further test the quality of our CLASH data, we focussed on the 24 experimentally verified sRNA-mRNA interactions recovered in our data, which we used as a 'ground truth' for known inter-actions. Strikingly, 92% of the sRNAs in our chimeras with experimentally verified interactions were fused to the cognate mRNA fragments (***Figure 2—figure supplement 1A***). Vice versa, ~87% of the mRNAs in our chimeras known to be regulated by sRNAs, were fused to cognate sRNA fragments (***Figure 2—figure supplement 1B***). Except for the GcvB-*sstT* chimeras, all the experimentally veri-fied interactions in our data had the known mRNA and sRNA seeds (***Figure 2—figure supplement 1C–D***). This implies that the false negative rate in our data is very low. When we extended these analyses to *all* sRNAs and mRNAs identified in our data, we obtained very similar results (***Figure 2—figure supplement 2A–B***). Only the known MicC seed sequence was absent in MicC chimeras (***Figure 2—figure supplement 2C***).

As a proxy for noise we quantified intermolecular chimeras containing rRNA sequences. Ribo-somal RNA represents up to 80% of total cellular RNA and therefore often contributes significantly to noise in sequencing data. Although Hfq is known to interact with rRNA, this interaction appears to be sRNA independent (***Andrade et al., 2018***). Therefore, chimeras containing rRNA fragments likely represent background. In less than 4% of the chimeras were sRNAs or mRNAs fused to rRNA sequences, suggesting that the CLASH data has low background (***Figure 2—figure supplements 1–2***).

We recovered around 20% of the sRNA-mRNA networks found with RIL-seq (***Figure 2B***) and 37 experimentally verified interactions (***Supplementary file 7***). These results suggest that while the CLASH data contained many known interactions, the analyses were clearly not exhaustive (also see Discussion). A large number of sRNA-mRNA interactions (~1700) were uniquely found in the CLASH data (***Figure 2B***) and many were supported by a relatively low number of reads compared to those found both in RIL-seq and CLASH (***Supplementary file 2***; ***Figure 2—figure supplement 3***). This raises the question whether these chimeras represent bona fide interactions or were merely gener-ated through random/stochastic ligation events. To address this, we repeated the previous bioinfor-matics analyses on the chimeras unique to the CLASH data. This gave almost identical results. The vast majority of the chimeras were fusions between sRNA and mRNA fragments (***Figure 2—figure supplement 4A–B***) and again in almost all cases the experimentally verified sRNA seeds were recov-ered (***Figure 2—figure supplement 4B***). Next, we analysed the chimeras unique to the CLASH data that were supported by less than four reads (***Figure 2—figure supplement 5***). The majority of these chimeras in this group represented sRNA-mRNA and mRNA-mRNA interactions (***Figure 2—figure supplement 5A–B***) and again in almost all cases the known sRNA seed sequences were recovered (***Figure 2—figure supplement 5C***). We do note the slightly higher percentage of sRNA-rRNA and mRNA-rRNA chimeras (12–13%) in this group, suggesting higher background levels (***Figure 2—fig-ure supplement 5A–B***). However, considering again the sheer abundance of rRNA in bacterial cells, we argue that also the background in this group of low abundance chimeras is remarkably low.

To provide additional evidence that the low abundant interactions identified with CLASH represent genuine interactions and not weak or stochastic interactions, we calculated the base-pairing potential between the two halves of the chimeras. For this purpose, we used RNAduplex (*Lorenz et al., 2011*) to compute the hybridization potential (in kcal/mol) of the two halves in each chimera. We focussed on sRNA-mRNA chimeras as this group represented the largest number of interactions (*Figure 3*). These analyses revealed that all sRNA-mRNA chimeras in the CLASH data, even those supported by only a few reads (*Figure 3D*), had a significantly higher propensity to form stable duplexes when compared to in silico shuffled chimeric reads (p-value<$6*10^{-16}$). These data imply that a large fraction of the chimeras represent genuine base-pairing interactions and not random ligation events.

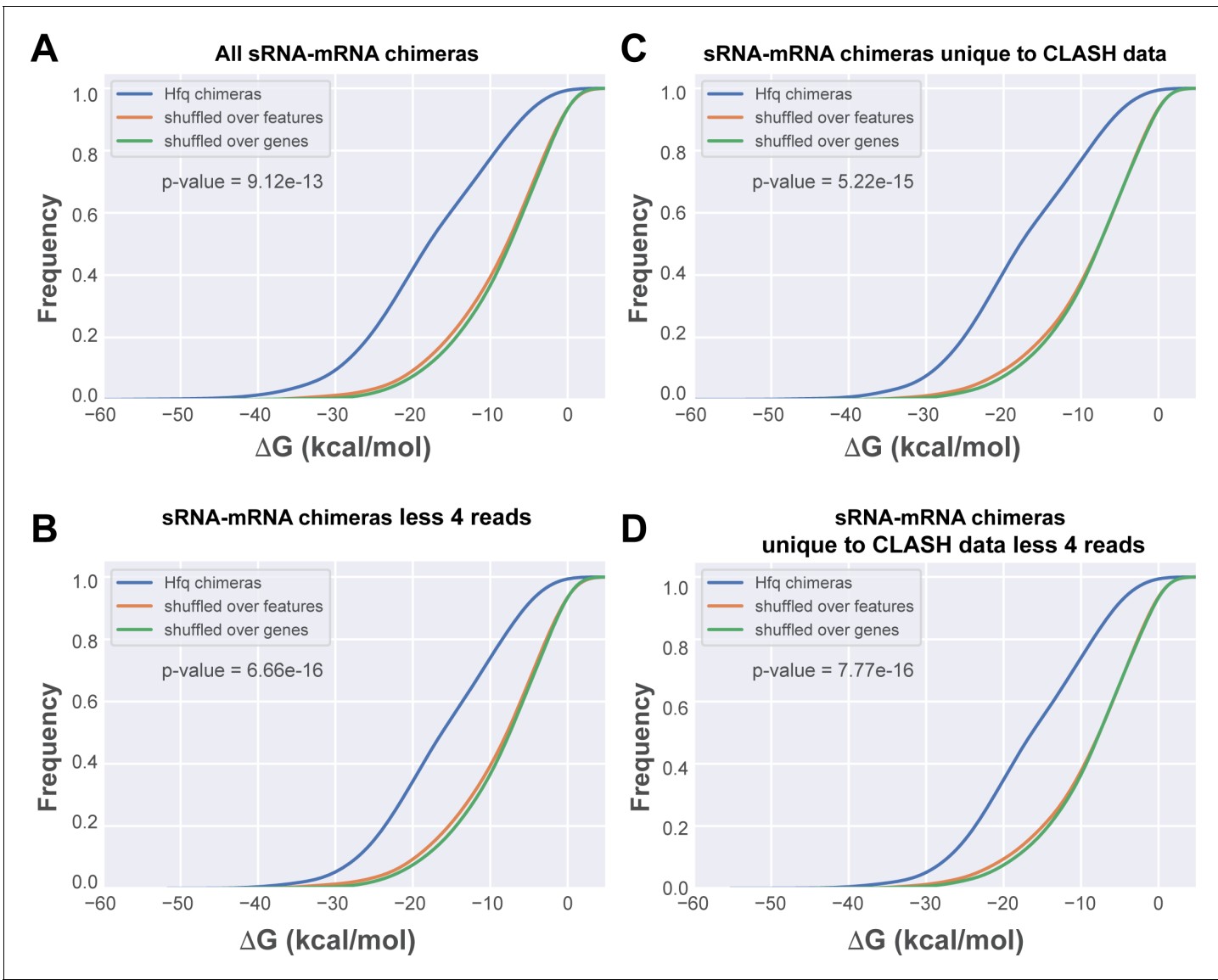

**Figure 3.** In silico folding of sRNA-mRNA chimeras shows Hfq CLASH sRNA-mRNA interactions are significantly more structured than randomly matched pairs. (A) Cumulative distribution of the predicted folding energy (ΔG) values between sRNA and matching mRNA found in all statistically filtered sRNA-mRNA interactions. Chimera folding energies were calculated using RNADuplex (*Lorenz et al., 2011*), and their distribution was compared to the control distributions of chimeric reads in which the fragments were randomly shuffled over the same gene, or over genes belonging to the same class of genes (e.g sRNAs or mRNAs), respectively. Significance was tested with Kolmgorov-Smirnov test. (B) As in (A) but now for the chimeras unique to the CLASH data. (C) As in (A) but now for chimeras that are supported by less than four reads. (D) As in (A) but now for chimeras unique to the CLASH data and supported by less than four reads.

If the recovered interactions indeed represent bona fide interactions, then it may be expected that the putative mRNA targets found in CLASH chimeras are enriched for sequence motifs complementary to the sRNA seed sequences. To test this, we performed motif analyses on targets of 38 sRNAs that showed at least five unique interactions with different mRNAs (*Figure 4A*). Some sRNAs appeared to utilize multiple and independent seed sequences to base-pair with mRNAs. In these cases, we first performed a K-means clustering analysis to group those chimeras that contained similar sRNA sequences. For each of the resulting clusters (usually 4–5), we subsequently extracted the corresponding mRNA fragments from the chimeras and performed motif analyses using the MEME tool suite (*Bailey et al., 2009*). This enabled us to detect mRNA sequence motifs that are associated with specific sRNA seed sequences. The results are shown in *Figure 4—figure supplements 1–12*. The motif analyses were performed for all the mRNA fragments found in sRNA-mRNA chimeras, mRNA fragments from sRNA-mRNA interactions uniquely identified by CLASH, and mRNA fragments found in sRNA-mRNA interactions supported by less than four reads. In the majority of cases we recovered previously identified mRNA sequence motifs (*Faigenbaum-Romm et al., 2020*; *Melamed et al., 2016*; *Waters et al., 2017*). The majority of the sRNA-mRNA interactions involving RyjB, ChiX, SdsR and GadY were supported by less than four reads and only found in our CLASH data. Regardless, the mRNA fragments in these chimeras were significantly enriched for sequence motifs complementary to the sRNA including known seed sequences (*Figure 4B*, *Figure 4—figure supplements 1–3*). We also identified novel mRNA sequence motifs for RyjB, GadY, ArcZ, CyaR and GcvB (*Figure 4B*, *Figure 4—figure supplements 3–6*). GcvB was previously reported to recognize the consensus motif CACAaCAY in mRNAs through interactions with the GU-rich R1 seed region located at bases 66–89 (*Gulliver et al., 2018*; *Sharma et al., 2011*). Consistent with this, we found a similar motif in chimeras from cluster 2, although these were less frequently recovered in the interactions only identified by CLASH and chimeras supported by less than four reads. Our analyses also identified a well-defined sequence motif in putative mRNA targets that is highly complementary to the R3 seed, consistent with the idea that this seed is also very frequently used to regulate mRNAs (*Lalaouna et al., 2019*). The R3 complementary sequence motif was most highly enriched in the interactions uniquely identified in CLASH (*Figure 4—figure supplement 6B*). In all but one case (CyaR motif in cluster 3; *Figure 4—figure supplement 5B*) did the mRNA sequence motifs show significant complementarity to known seed sequences (*Figure 4—figure supplements 1–12*). In addition, these analyses indicated that sequences in the 3' ends of ArcZ and CyaR can also function as seeds (*Figure 4—figure supplements 4–5*). Certain motifs were more frequently found in sRNA-mRNA interactions uniquely identified by CLASH: The MgrR mRNA motif found in the RIL-seq data was not frequently detected in our data, but the novel MgrR interactions recovered by CLASH showed a significant enrichment of G-rich motifs in mRNA fragments (*Figure 4—figure supplement 7*).

We also reasoned that genuine interactions should be enriched in RNA-RNA interaction data generated by alternative experimental approaches. To test this, we compared our data to recent GcvB and CyaR MS2 Affinity Purification coupled with RNA Sequencing (MAPS) datasets (*Lalaouna et al., 2019*; *Lalaouna et al., 2018*; *Figure 4—figure supplement 13A and B*). The CyaR and GcvB datasets were chosen as we had a large number of different mRNA interactions with these sRNAs (>200), which enabled us to do a statistically meaningful comparison of the datasets. Indeed, the results show that CLASH mRNA targets were significantly more highly enriched compared to the other genes in the MAPS datasets. This was even the case for those interactions supported by a relatively low number of chimeric reads, including many interactions uniquely found in our CLASH data.

Collectively, these analyses strongly suggest that the predicted interactions found in our CLASH data, even those supported by a relatively low number of chimeras, are highly enriched for *bona fide* sRNA-mRNA interactions and less likely to be formed by random/stochastic events.

What is the biological significance of these interactions? Because sRNAs can influence the stability of their mRNA targets, we asked how many of the putative mRNA targets showed changes in gene expression in existing sRNA over-expression datasets (*Figure 5*, *Figure 5—figure supplements 1–4*). We initially analysed previously published *E. coli* microarray datasets (*Beisel and Storz, 2011*; *De Lay and Gottesman, 2009*; *Sharma et al., 2011*) similar to what was performed to validate RIL-seq interactions (*Melamed et al., 2016*). For these analyses, we also focussed our analyses on sRNAs that had a very high number of different mRNA interactions (>200) in our CLASH data (ArcZ, GcvB, CyaR and Spot42; *Figure 5—figure supplements 1–4*). While this work was under revision, RNA-

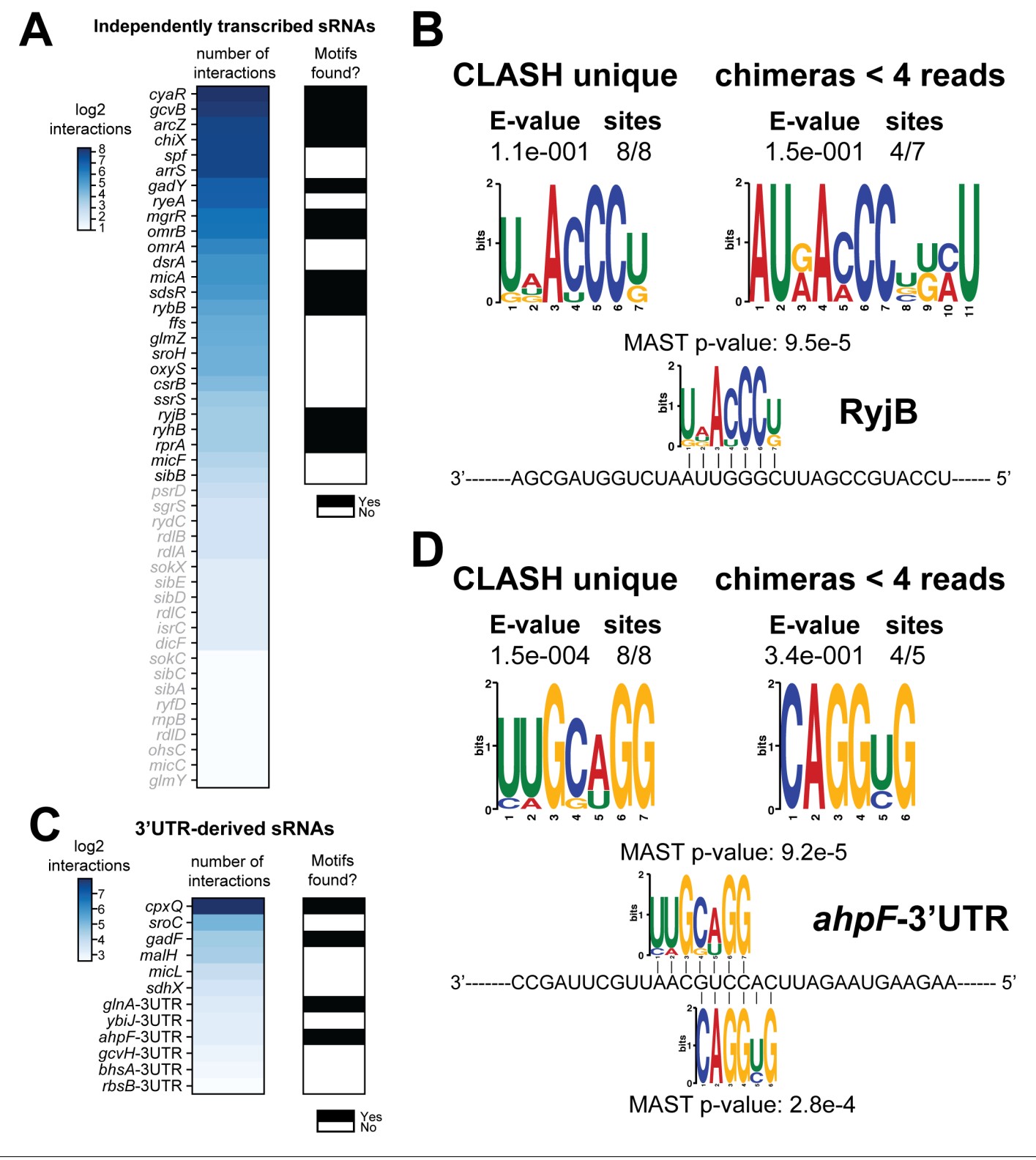

**Figure 4.** Total number of interactions for sRNAs and in how many cases enriched sequence motifs were found. (**A** and C) The heatmaps show the number of different mRNA interactions identified with independently transcribed sRNAs (**A**) or (putative) 3'UTR-derived sRNAs (**C**). Only the sRNA for which we recovered at least five different interactions with mRNAs (highlighted in black) were further analysed for enriched motifs in the putative mRNA targets. The black-and-white heatmaps indicate if enriched motifs were identified in predicted mRNA targets (black means Yes and white means No).

*Figure 4 continued on next page*

*Figure 4 continued*

Motif analysis was performed using the MEME suite (*Bailey et al., 2009*). The number of target sequences that contained the common motif and the E-value of MEME are shown. The identified motifs in the mRNA targets also show sequence complementarity to the sRNA sequence. The Motif Alignment Search Tool (MAST) was used to determine the degree of complementarity between the identified motifs in putative mRNA targets and the putative sRNA. An sRNA was considered to have an enriched motif if a motif identified by MEME had an E-value <= 0.1 and/or the MAST p-value of the motif, which indicates the overall match between the identified motifs and the sRNA sequence (*Bailey et al., 2009*), was <= 0.001. (B–D) Motif analyses of mRNA sequences found in RyjB sRNA-mRNA and *ahpF*-3'UTR-mRNA interactions. All of the RyjB and *ahpF-3'UTR* interactions with mRNAs we found were uniquely detected in our CLASH data.

The online version of this article includes the following figure supplement(s) for figure 4:

**Figure supplement 1.** Identification of complementary sequence motifs in predicted ChiX mRNA targets.
**Figure supplement 2.** Identification of complementary sequence motifs in predicted SdsR mRNA targets.
**Figure supplement 3.** Identification of complementary sequence motifs in predicted GadY mRNA targets.
**Figure supplement 4.** Identification of complementary sequence motifs in predicted ArcZ mRNA targets.
**Figure supplement 5.** Identification of complementary sequence motifs in predicted GadY mRNA targets.
**Figure supplement 6.** Identification of complementary sequence motifs in predicted GcvB mRNA targets.
**Figure supplement 7.** Identification of complementary sequence motifs in predicted MgrR mRNA targets.
**Figure supplement 8.** Identification of complementary sequence motifs in predicted MicA mRNA targets.
**Figure supplement 9.** Identification of complementary sequence motifs in predicted RybB mRNA targets.
**Figure supplement 10.** Identification of complementary sequence motifs in predicted OmrB mRNA targets.
**Figure supplement 11.** Identification of complementary sequence motifs in predicted RyhB mRNA targets.
**Figure supplement 12.** Identification of complementary sequence motifs in predicted RprA mRNA targets.
**Figure supplement 13.** CLASH targets are highly enriched in MAPS data.

seq data from several sRNA over-expression analyses in *E. coli* became available (*Faigenbaum-Romm et al., 2020*), which we subsequently included in our analyses (*Figure 5A*). Only a subset of the predicted sRNA targets showed significant changes in gene expression. GcvB CLASH mRNA targets were most highly enriched for differentially expressed genes, although this was lower for the less abundant interactions uniquely found in the CLASH data (*Figure 5A*, *Figure 5—figure supplement 1*). Surprisingly, although the CyaR targets were highly enriched in the MAPS data, only a few of the mRNAs were significantly differentially expressed in the CyaR over-expression data (*Figure 5A*, *Figure 5—figure supplement 2*). The Spot42 mRNA targets predicted by CLASH showed larger (albeit modest) changes in gene expression compared to the other genes in the dataset (*Figure 5—figure supplement 3*).

Previous work implied that those interactions that impact mRNA steady-state levels are mostly found in multiple replicate RIL-seq experiments and are generally more abundant (*Faigenbaum-Romm et al., 2020*). The interactions recovered by both RIL-seq and CLASH were supported by a significantly higher number of chimeras compared to those uniquely identified in the CLASH data (*Figure 2—figure supplement 3*). Therefore, we asked if this group of interactions was more likely to alter mRNA levels. This was the case for the GcvB and MicA mRNA interactions, but not ArcZ and CyaR interactions (*Figure 5B*).

In conclusion, similar to what was observed for RIL-seq mRNA targets (*Faigenbaum-Romm et al., 2020*), many of the sRNA-mRNA interactions do not appear to significantly affect mRNA steady-state levels and for some sRNAs reproducible interactions have a higher likelihood impacting mRNA target levels (also see Discussion).

## Hfq CLASH predicts sRNA-sRNA interactions as a widespread layer of post-transcriptional regulation

Surprisingly, we uncovered a large number of sRNA-sRNA chimeras, representing 200 unique interactions (*Figure 2A*; 2.1%; *Supplementary file 4*). Many of the sRNA-sRNA interactions were uniquely found in our Hfq CLASH data (*Figure 6A*), were growth-stage specific and the sRNA-sRNA networks show extensive rewiring across the exponential, transition and stationary phases (*Figure 6—figure supplement 1*). The sRNA-sRNA network is dominated by several abundant sRNAs that appear to act as hubs with many interacting partners: ChiX, Spot42 (spf), ArcZ and GcvB. Again, in many cases, the experimentally validated sRNA seed sequences were found in the chimeric reads, for both established and novel interactions. For example, the majority of ArcZ sRNA-sRNA chimeras

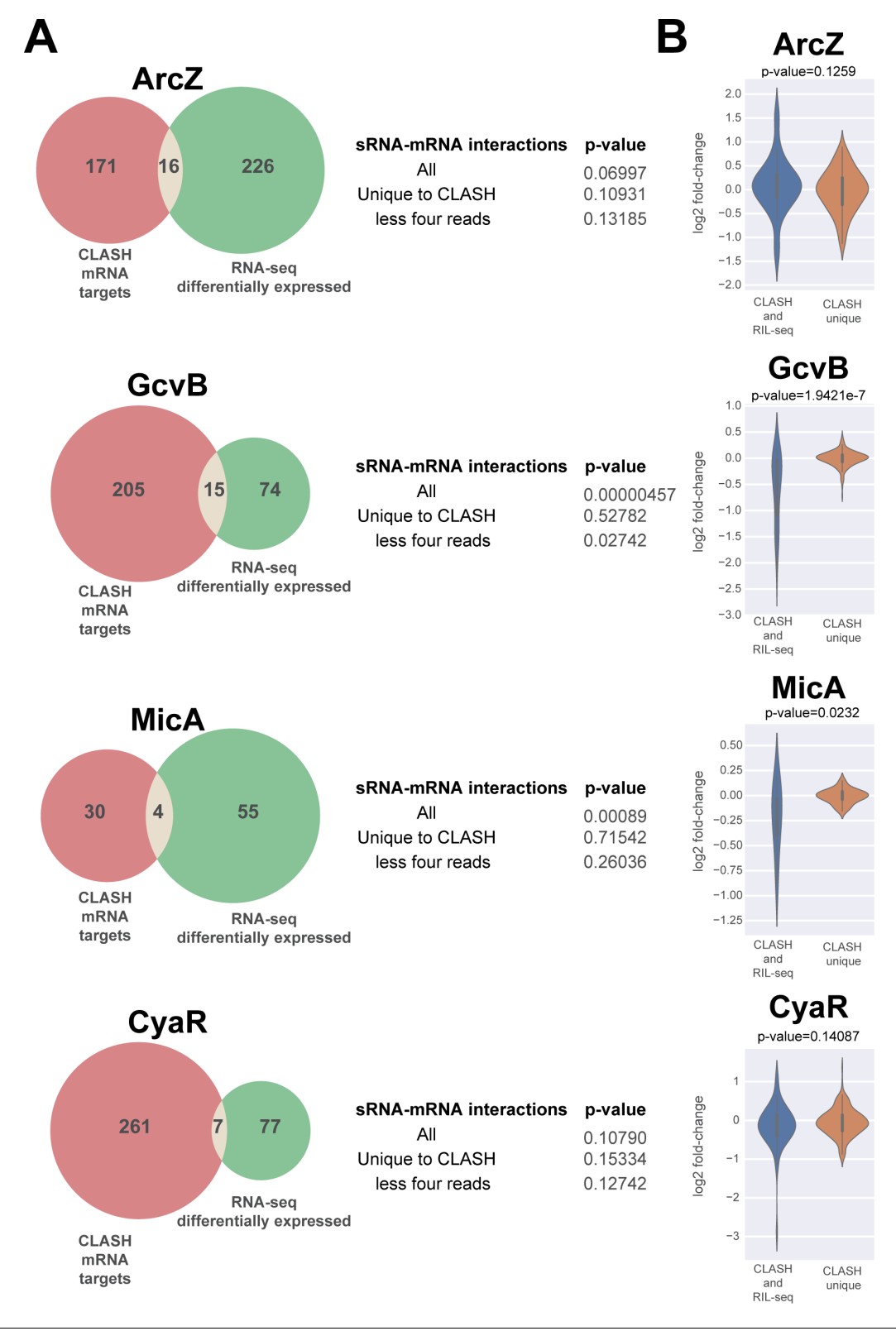

**Figure 5.** A subset of putative mRNA targets identified by CLASH show gene expression changes upon over-expression of the sRNA. The Venn diagrams show how many of the predicted mRNA targets were also found to be differentially expressed in sRNA over-expression RNA-seq data (*Faigenbaum-Romm et al., 2020*). The GcvB and MicA CLASH mRNA targets are highly enriched for genes that are differentially expressed in the over-expression RNA-seq data (p-value<0.001). The statistical significance was calculated using a hypergeometric test. Interactions that are generally

*Figure 5 continued on next page*

Figure 5 continued

presented by a relatively low number of reads ('CLASH unique' and 'less four reads' categories) are not significantly enriched for differentially expressed genes. (B) The mRNA targets found in GcvB and MicA interactions found in both RIL-seq and CLASH show significantly higher fold-changes in the over-expression data compared to the interactions uniquely found in the CLASH data. The violin plots show the distribution of fold-changes in mRNA target expression (y-axis) in the over-expression RNA-seq data for chimeras supported by CLASH and RIL-seq and those found in CLASH only (x-axis). Statistical significance between the two groups was calculated using a Mann-Whitney U test.

The online version of this article includes the following figure supplement(s) for figure 5:

**Figure supplement 1.** Impact of the identified interactions on gene expression levels of GcvB mRNA targets predicted by CLASH.
**Figure supplement 2.** Impact of the identified interactions on gene expression levels of CyaR mRNA targets predicted by CLASH.
**Figure supplement 3.** Impact of the identified interactions on gene expression levels of Spot42 mRNA targets predicted by CLASH.
**Figure supplement 4.** Impact of the identified interactions on gene expression levels of ArcZ mRNA targets predicted by CLASH.

contained the known and well conserved seed sequence (*Figure 6B*, *Figure 6—figure supplement 2*).

The sRNA-sRNA chimeras containing CyaR fragments were of particular interest, as the sRNA is primarily expressed during the transition from late exponential to stationary phase (*De Lay and Gottesman, 2009*). While 30% of the CyaR chimeras contained the known seed sequence (*De Lay and Gottesman, 2009*; *Papenfort et al., 2008*), the majority of the chimeras contained a ~25 nt fragment in the 5' region of CyaR, which was also frequently recovered in RNase E CLASH data (*Waters et al., 2017*; *Figure 6B*; *Figure 6—figure supplement 2*), suggesting that this region represents a *bona fide* interaction site. Notably, the ArcZ-CyaR chimeras contained the seed sequences from both sRNAs (*Figure 6—figure supplement 2*) and these were detected specifically in the transition phase (*Figure 6A*; *Figure 6—figure supplement 1*).

To validate the predicted in vivo interaction between ArcZ and CyaR (*Figure 7A*), we used an *E. coli* plasmid-based assay that is routinely used to monitor sRNA-sRNA interactions and expression of their target mRNAs (*Melamed et al., 2016*; *Miyakoshi et al., 2015b*; *Tree et al., 2014*). An advantage of this system is that each sRNA would be uncoupled from the chromosomally encoded regulatory networks (that were thought to act largely in a 1:1 stoichiometry) and to allow the specific effects of the sRNA-target RNA to be assessed (*Miyakoshi et al., 2015b*). Importantly, these sRNAs were induced during early exponential growth phase when the endogenous (processed) ArcZ and CyaR sRNAs are only detectable at very low levels (*Figure 7B*, lanes 1, 2, 5, 7). The RT-qPCR data for each pZE-expressed sRNA were subsequently normalized to the results obtained with the pJV300 control to calculate fold changes in expression levels. It has recently been shown that sRNAs can also function as 'decoys' or 'sponges' that can divert other sRNA away from its mRNA targets (*Azam and Vanderpool, 2015*; *Figueroa-Bossi and Bossi, 2018*; *Kavita et al., 2018*). This mode of 'regulating the regulator' often results in cross-talk between pathways (reviewed in *Figueroa-Bossi and Bossi, 2018*). We hypothesized that the ArcZ-CyaR interaction may represent such a sponging activity. However, since it is difficult to predict directly from the CLASH data which sRNA in each pair acts as the decoy/sponge, we tested both directions. ArcZ over-expression not only decreased the expression of its mRNA targets (*tpx*, *sdaC*) by more than 50%, but also that of CyaR (*Figure 7C*, panel I; *Figure 7D*, panel I). Concomitantly, we observed a substantial increase in CyaR targets *nadE* and *yqaE* (*Figure 7C*, panel I). CyaR over-expression reduced the level of a direct mRNA target (*nadE*) by ~40% but it did not significantly alter the level of ArcZ or ArcZ mRNA targets (*tpx* and *sdaC*; *Figure 7C*, panel II). Notably, in this two-plasmid assay CyaR was not expressed at levels higher than ArcZ (*Figure 7D*, panel II). Therefore, it is plausible that under the tested conditions the CyaR over-expression was not sufficient to see an effect on ArcZ. We find this unlikely as over-expression of CyaR also did not significantly affect endogenous ArcZ levels, which was ~80 fold less abundant than CyaR in this experiment (*Figure 7D*, panel III). The qPCR results were also confirmed by Northern blot analyses (*Figure 7B*, lanes 1–8), which confirmed the reduction in CyaR levels upon ArcZ over expression and demonstrated that ArcZ processing was not affected upon CyaR over-expression. These results suggest that the regulation is unidirectional, reminiscent of what has been described for Qrr3 in *Vibrio harveyi* (*Feng et al., 2015*).

To provide additional support for direct interactions between these sRNAs, we generated mutations in the seed sequences of the sRNAs analysed here (*Figure 7A*). We found that two G to C nucleotide substitutions in ArcZ was sufficient to disrupt ArcZ regulation of CyaR (*Figure 7C* panel

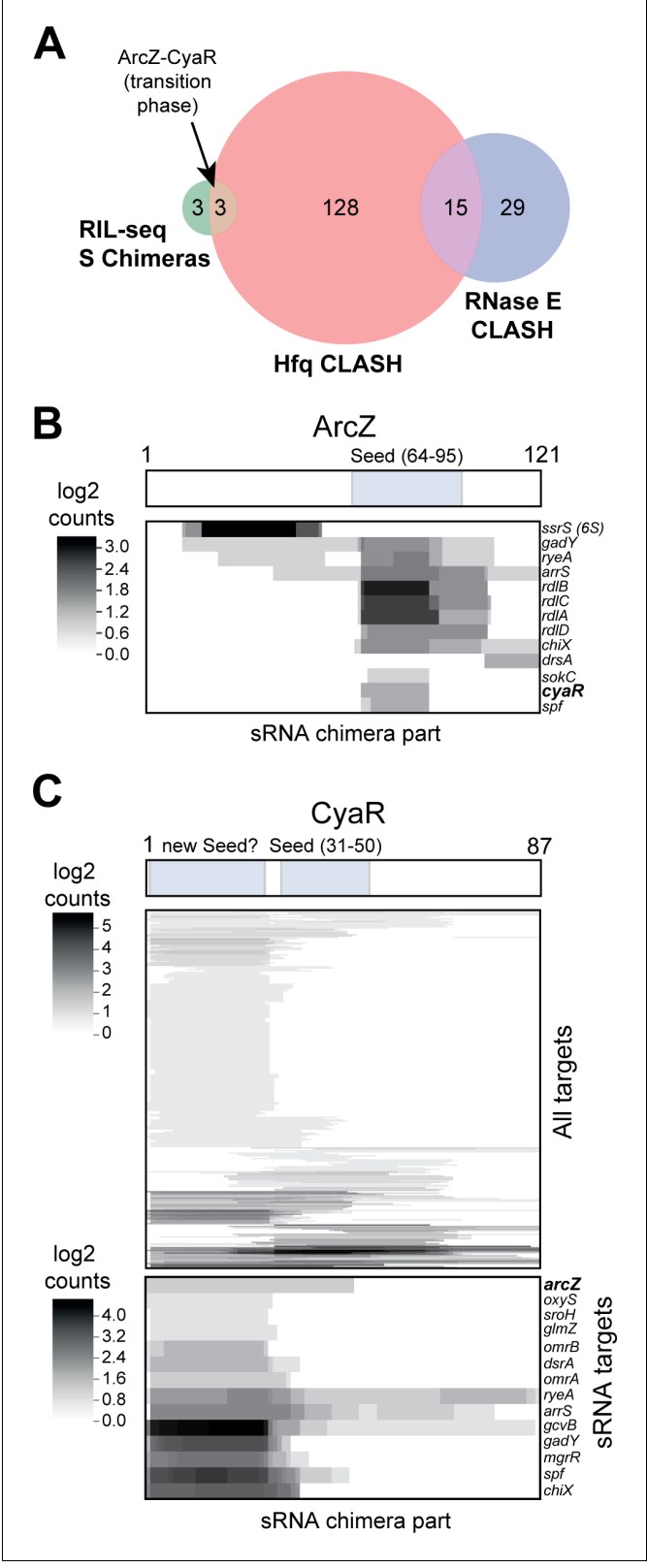

**Figure 6.** sRNA-RNA interactions identified by CLASH. (**A**) Hfq CLASH uncovers sRNA-sRNA interaction networks: comparison between statistically filtered sRNA-sRNA interactions in the Hfq CLASH data, RIL-seq S-chimeras (*Melamed et al., 2016*) (log and stationary) and RNase E CLASH (*Waters et al., 2017*). Only independently transcribed sRNAs were considered. (**B–C**) Heatmaps showing the read density (log$_2$(chimera count

*Figure 6 continued on next page*

*Figure 6 continued*

+1)) of chimeric fragments mapping to ArcZ (**B**) and CyaR (**C**). The location of the known sRNA seed sequences as well as the predicted new CyaR seed is indicated above the heatmap. Note that the ArcZ processing site is located just upstream of the seed sequence.

The online version of this article includes the following figure supplement(s) for figure 6:

**Figure supplement 1.** sRNA-RNA interactions identified by CLASH are growth-stage specific.

**Figure supplement 2.** Interactions between ArcZ, CyaR and GcvB are conserved.

---

III; ArcZ 70–71 + CyaR). Unexpectedly, the wild-type ArcZ was also able to effectively suppress the CyaR seed mutant (*Figure 7C* panel III; ArcZ + CyaR 38–39). We predict that the wild-type ArcZ can still form stable base-pairing interactions with the CyaR mutant. Nevertheless, regulation by the ArcZ 70–71 mutant was almost fully restored when complementary mutations were introduced in the CyaR region (*Figure 7C* panel III; ArcZ 70–71 + CyaR 38–39), providing additional evidence that these sRNAs base-pair in vivo. Furthermore, the data also demonstrate that it is very unlikely that the observed changes in CyaR levels were the result of Hfq redistribution due to over-expression of ArcZ (*Moon and Gottesman, 2011*; *Papenfort et al., 2009*), as the ArcZ seed mutant stably accumulated (and therefore effectively binds Hfq), but did not affect CyaR levels (*Figure 7C* panel III).

These results, together with the CLASH data, imply that ArcZ and CyaR base-pair in vivo, and that this interaction could lead to a reduction in CyaR levels but not vice versa.

## Hfq CLASH identifies novel sRNAs in untranslated regions

Two lines of evidence from our data indicate that many other mRNAs may be harbouring sRNAs in their UTRs or be involved in base-pairing among themselves. First, around 10% of the unique intermolecular chimeras mapped to mRNA-mRNA interactions (*Figure 2A*). Secondly, we observed extensive binding of Hfq in 3'UTRs near transcriptional terminators (*Figure 1—figure supplement 3A–B*), indicating that like in *Salmonella*, the *E. coli* 3'UTRs may harbour many functional sRNAs (*Chao et al., 2017*). We identified 116 3'UTR-containing mRNA fragments that were involved in 507 interactions (represented by a total of 3149 unique chimeras). Eighteen of these 3'UTR fragments were also identified in 3'UTR-mRNA chimeric reads in the RIL-seq S-chimeras data (*Melamed et al., 2016*) and 10 appeared stabilised upon transient inactivation of RNase E performed in *Salmonella* (TIER-seq data *Chao et al., 2017*; *Figure 8A*, *Supplementary files 5* and *6*). For several of the putative 3'-UTR derived sRNAs, complementary sequence motifs in the mRNA fragments were identified, including motifs for the putative sRNA derived from the 3'UTR of *ahpF* (*Figure 4C–D*; *Figure 8—figure supplements 1–3*). Out of the 507 3'UTR-mRNA interactions, 75 were 3'UTRs fused to 5'UTRs of mRNAs, suggesting that these may represent 3'UTR-derived sRNAs that base-pair with 5'UTRs of mRNAs, a region frequently targeted by sRNAs (*Supplementary files 5* and *6*). Strikingly, 233 interactions (2094 unique chimeras) contained the 3'UTR fragment of *cpxP*, 51 (812 chimeras) of which were also found in the RIL-Seq data (*Supplementary file 6*). In *Salmonella cpxP* harbours the CpxQ sRNA (*Chao and Vogel, 2016*). Our analyses greatly increased the number of potential CpxQ mRNA targets and show that the vast majority of CpxQ interactions take place during the transition and stationary phases (*Supplementary file 6*). Motif analyses of the putative CpxQ mRNA targets, including those identified in the interactions unique to CLASH, revealed two highly enriched G-rich sequence motifs that showed strong sequence complementarity to the known seed sequences (*Figure 8—figure supplement 2*).

We identified six mRNA 3'UTRs that were uncovered in all three (Hfq CLASH, RIL-seq and TIER-seq) datasets (*Figure 8A*), suggesting they likely contain sRNAs released from 3'UTRs by RNase E processing. Northern blot analyses confirmed the presence of sRNAs in *malG*, *ygaM* and *gadE* 3'UTRs (*Figure 8B*, *Figure 8—figure supplement 4*). We predict that the 3'UTR of *ygaM* harbours a ~100 nt sRNA (hereafter referred to as YgaN; *Figure 8—figure supplement 4*) and robust Hfq cross-linking could be detected in this region (*Figure 8C*).

The *gadE* 3'UTR was also detected in the RIL-seq data and experimentally confirmed and annotated as GadF (*Melamed et al., 2016*). Remarkably, even though we only recovered 23 unique GadF-mRNA interactions, two distinct complementary sequence motifs (CCAGGGG and CUGGUG) were identified in mRNA fragments of these chimeras, the former of which was not previously

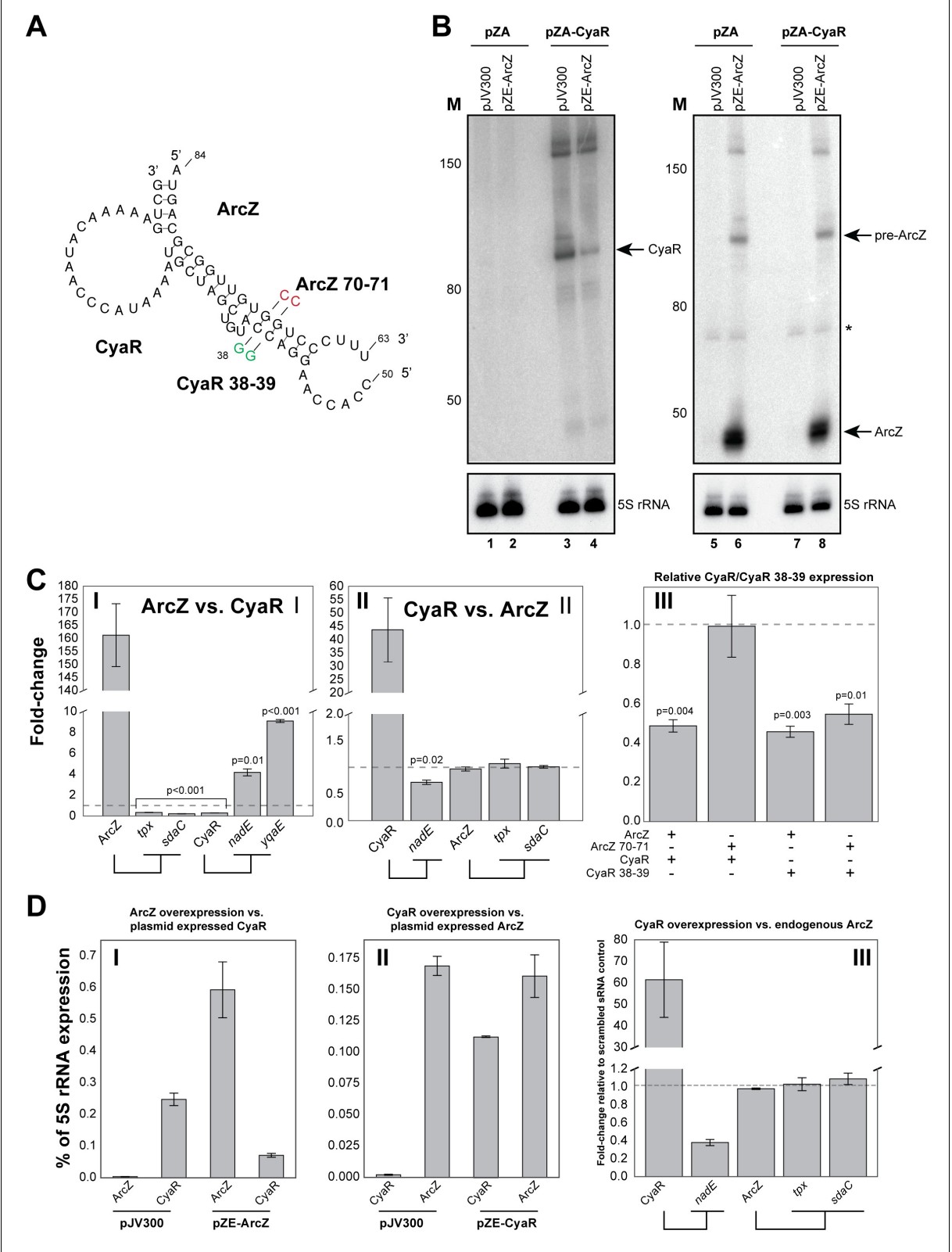

**Figure 7.** ArcZ can influence CyaR levels. (**A**) Base-pairing interactions predicted from the ArcZ-CyaR chimeras using RNACofold. The nucleotide substitutions for experimental validation of direct base-pairing are shown as red or green residues. (**B**) Northern blot analysis of ArcZ and CyaR. The cells containing both the empty pZA and pJV300 plasmids (lanes 1, 5, 7) do not express ArcZ and CyaR at detectable levels. (**C**) Validation of ArcZ-CyaR interaction by over-expression analyses. ArcZ (panel I) orCyaR (panel II) was over-expressed and the levels of their targets were monitored by RT-qPCR.

*Figure 7 continued on next page*

*Figure 7 continued*

The *tpx* and *sdaC* mRNAs are ArcZ targets (panel I). The *nadE* and *yqaE* mRNAs are CyaR targets (panel II). The dashed horizontal line indicates the level in the control plasmid (pJV300) that expresses a ~50 nt randomly generated RNA sequence. Panel III: The sRNAs and mutants (as in (A)) were ectopically co-expressed in *E. coli* and CyaR and CyaR 38–39 levels were quantified by RT-qPCR. Experiments were performed in biological and technical triplicates; Error bars indicate the standard error of the mean (SEM) of the three biological replicates. (D) ArcZ and CyaR were overexpressed from a plasmid-borne IPTG inducible promoter (pZE-ArcZ and pZE-CyaR) and the data were compared to data from cells carrying plasmid pJV300. The co-expressed candidate target sRNAs (expressed from pZA-derived backbone) were induced with anhydrotetracycline hydrochloride (panels I and II). The bars indicate the mean fold-change in expression relative to the level of 5S rRNA (*rrfD*) in cells with the indicated vector. In panel III endogenous ArcZ levels were measured upon over-expression of CyaR. Error bars indicate the standard error of the mean from three biological replicates and three technical replicates per experiment. Source data are provided as a Source Data file.

The online version of this article includes the following source data for figure 7:

**Source data 1.** Source data for *Figure 7B*.
**Source data 2.** Source data for *Figure 7C*.
**Source data 3.** Source data for *Figure 7D*.

detected (*Figure 8—figure supplement 3*). Again, these complementary mRNA motifs were also enriched in interactions uniquely identified by CLASH (*Figure 8—figure supplement 3*). For two other 3'UTR-derived sRNAs (MicL and SdhX), we recovered 13 and 9 interactions with mRNAs, respectively (*Figure 8—figure supplement 5*). MicL was previously shown to repress the synthesis of the Lpp outer membrane protein (*Guo et al., 2014*). *Lpp* mRNA fragments were most frequently found in MicL chimeras (15; *Figure 8—figure supplement 5A*). The in silico folded structure of the MicL-*lpp* chimeras is in excellent agreement with the previously proposed interaction between MicL and *lpp* (*Figure 8—figure supplement 5B*; *Guo et al., 2014*). SdhX is involved in linking acetate metabolism with the TCA cycle (*De Mets et al., 2019*; *Miyakoshi et al., 2018*). Our data predict over a dozen SdhX interactions, several of which had not been previously described (*Figure 8—figure supplement 5C*). We recovered two SdhX interactions with known mRNAs targets (*ackA* and *katG*; *Figure 8—figure supplement 5D*; *De Mets et al., 2019*; *Miyakoshi et al., 2018*). Interestingly, the SdhX-*ackA* interaction was detected in the exponential phase, whereas the SdhX-*katG* interaction appeared specifically during stationary phase. Although the number of chimeras supporting these interactions were relatively low (*katG*; two chimeras; *ackA*; three chimeras), the in silico predicted interactions between the two halves of these chimeras are fully consistent with previously published work (*De Mets et al., 2019*; *Miyakoshi et al., 2018*). These results reinforce the idea that Hfq CLASH recovers genuine interactions.

To substantiate our 3'UTR-derived sRNA candidate prediction, we analysed RNA-seq data from a study that used Terminator 5'-Phosphate Dependent Exonuclease (TEX) to map transcription start sites (TSS) of coding and non-coding RNAs in *E. coli* (*Thomason et al., 2015*). TEX degrades processed transcripts that have 5' monophosphates, but not primary transcripts with 5' triphosphates. Therefore, these data enabled us to determine whether (a) a short RNA was detected in the 3'UTR and (b) whether these were generated by RNase-dependent processing (TEX sensitive) or originated from an independent promoter (TEX insensitive). For 47 of the 126 predicted 3'UTR-derived sRNAswe found strong evidence for the presence of sRNAs in the TEX data (*Figure 8—figure supplement 6*, *Supplementary file 5* and see Data and Code availability). The TEX data indicate that *ygaM* has (at least) two promoters, one of which is located near the 3' end of the gene that we predict is the TSS for YgaN (*Figure 8—figure supplement 6A*). Furthermore, we speculate that YgaN is processed by RNases. This is based on the observation that multiple YgaN species were detected in the Northern blot analyses (*Figure 8—figure supplement 4*) and the TEX data indicate that shorter YgaN RNAs are sensitive to TEX treatment (*Figure 8—figure supplement 6A*).

The majority of the sRNAs we analysed are more abundant at higher cell densities (including GadF, YgaN and RybB; see *Figure 8B*). In sharp contrast, the sRNA derived from the 3'UTR of the *malG* mRNA (MalH) was expressed very transiently and peaked at an $OD_{600}$ of 1.8 (*Figure 8B*). We envisage that the particularly transient expression of this sRNA may be associated with a role in the adaptive responses triggered during transition from exponential to stationary phases of growth.

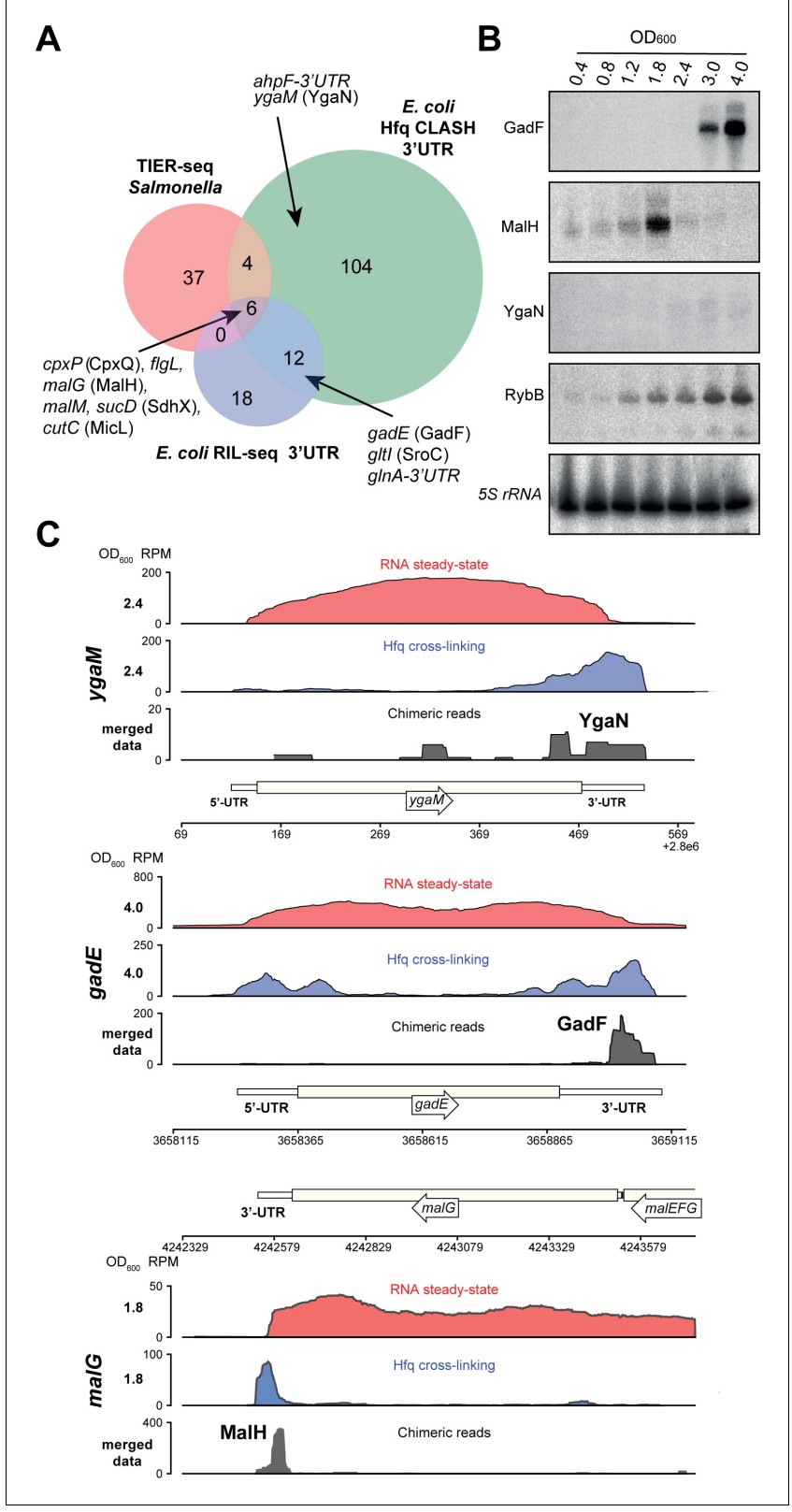

**Figure 8.** Hfq CLASH uncovers novel 3'UTR-derived sRNAs. (**A**) Genes with their 3'UTRs found fused to mRNAs were selected from the statistically filtered CLASH data and RIL-seq S-chimera data. The RIL-seq S-chimeras (**Melamed et al., 2016**) (log and stationary phases)were filtered for3'UTR/EST3UTR annotations on either orientation of the mRNA-mRNA pairs. Both were intersected with the set of mRNAs that were predicted by TIER-

*Figure 8 continued on next page*

*Figure 8 continued*

seq studies (*Chao et al., 2017*) to harbour sRNAs that get released from 3'UTRs by RNase E processing. Known (CpxQ, SdhX, MicL, GadF, *glnA*-3'UTR and SroC) and novel 3'UTR derived sRNAs (MalH, *flgL* 3'UTR, *ahpF*-3'UTR and YgaN) are indicated. See *Supplementary file 5* for the detailed comparison. (B) MalH is transiently expressed during the transition from exponential to stationary phase. RybB was probed as a sRNA positive control and 5S rRNA as the loading control. See *Figure 8—figure supplement 4* for full-size blots. (C) Genome-browser snapshots of several regions containing candidate sRNAs for optical densities at which the RNA steady-state was maximal; the mRNA names and $OD_{600}$ are indicated at the left side of the y-axes; the y-axis shows the normalized reads (RPM: reads per million); red: RPM of RNA steady-states from an RNA-seq experiment, blue: Hfq cross-linking from a CLASH experiment; black: unique chimeric reads found in this region.

The online version of this article includes the following figure supplement(s) for figure 8:

**Figure supplement 1.** Identification of complementary sequence motifs in predicted *glnA*-3'UTR mRNA targets.
**Figure supplement 2.** Identification of complementary sequence motifs in predicted CpxQ mRNA targets.
**Figure supplement 3.** Identification of complementary sequence motifs in predicted GadF mRNA targets.
**Figure supplement 4.** *YgaM*, *gadE* and *malG* contain sRNAs in their 3'UTRs.
**Figure supplement 5.** Hfq CLASH identifies known interactions between 3'-UTR derived sRNAs and mRNA targets.
**Figure supplement 6.** Analysis of exonuclease (TEX) RNA-seq datasets.

## Discussion

Microorganisms need to constantly adapt their transcriptional program to meet changes in their environment, such as changes in temperature, cell density and nutrient availability. In bacteria, small RNAs (sRNAs) and their associated RNA-binding proteins play a key role in this process. By controlling translation and degradation rates of mRNAs in response to stress (*Holmqvist and Wagner, 2017*; *Nitzan et al., 2017*; *Shimoni et al., 2007*), they can regulate the kinetics of gene expression as well as suppress noisy signals (*Beisel and Storz, 2011*), enabling organisms to more efficiently adapt to environmental changes. A major challenge for bacteria is the transition from exponential growth to stationary phase, when the most favourable nutrients become limiting. To counteract this challenge, cells need to rapidly remodel their transcriptome to efficiently metabolize alternative carbon sources. This transition is highly dynamic and involves both activation and repression of diverse metabolic pathways. However, it is unclear to what degree sRNAs contribute to this transition. The most useful piece of information would be to know what sRNAs are upregulated during this transition phase and to identify their RNA targets. This would help to uncover the regulatory networks that govern this adaptation, as well as provide a starting point for more detailed functional analyses on sRNAs predicted to play a key role in this process. For this purpose, we performed UV cross-linking, ligation and sequencing of hybrids (CLASH; *Kudla et al., 2011*) to unravel the sRNA base-pairing interactions during this transition. Using Hfq as a bait we uncovered thousands of unique sRNA base-pairing interactions. We identified almost 1700 novel sRNA-mRNA interactions represented by over 18000 unique chimeras, and 200 novel sRNA-sRNA interactions, compared to previously published work (*Melamed et al., 2016*; *Waters et al., 2017*).

### Hfq CLASH

Our earlier *S. cerevisiae* Cross-linking and cDNA analysis data (CRAC; *Granneman et al., 2009*) showed that a percentage of the cDNAs were formed by intermolecular ligations of two RNA fragments (chimeras) known to base pair in vivo (*Kudla et al., 2011*). These findings prompted us to develop a refined protocol to enrich for sRNA-target chimeric reads using Hfq as an obvious bait. The initial Hfq UV cross-linking data (CRAC; *Tree et al., 2014*) did not yield sufficiently high numbers of chimeric reads to extract new biological insights. In line with observations from other groups (*Bandyra et al., 2012*; *Bruce et al., 2018*; *Morita et al., 2005*), it was proposed that duplexes formed by Hfq are rapidly transferred to the RNA degradosome. This can cause an extensive reduction in the likelihood of capturing sRNA-target interactions with Hfq using CLASH (*Waters et al., 2017*). However, a recent study demonstrated that Hfq can be used effectively as a bait to enrich for sRNA-target duplexes under lower stringency purification conditions suggesting that sRNA-mRNA duplexes are sufficiently stable on Hfq during purification (*Melamed et al., 2016*). This encouraged us to further optimize the CLASH method. We made a number of changes to the protocol that

enabled us to recover a large number of chimeric reads, many of which represent sRNAs base-paired to potential targets (detailed in Materials and methods). We shortened various incubation steps to minimize RNA degradation and performed very long and stringent washes after bead incubation steps to remove any background binding of non-specific proteins and RNAs. Crucially, we very carefully controlled the RNase digestion step that is used to trim the cross-linked RNAs prior to making cDNA libraries, ensuring the recovery of longer chimeric RNA fragments. The resulting cDNA libraries were paired-end sequenced to increase the recovery of chimeric reads with high mapping scores from the raw sequencing data. These modifications led to a substantial improvement in the recovery of chimeric reads (8.6% compared to 0.001%; 0.47% were intermolecular chimeras).

Both RIL-seq and Hfq CLASH have advantages and disadvantages and are highly complementary. A major strength of CLASH, however, is that the purification steps are performed under highly stringent and denaturing conditions. During the first FLAG affinity purification steps, the beads are extensively washed with high-salt buffers and the second Nickel-affinity purification step is done under denaturing conditions (6M guanidium hydrochloride). These stringent purification steps can significantly reduce noise by strongly enriching for RNAs covalently cross-linked to the bait protein (*Granneman et al., 2009*). Indeed, we show that Hfq CLASH can generate high-quality RNA-RNA interaction data with low background: only a few hundred chimeric reads were found in the control datasets, compared to the over 50,000 chimeras that co-purified with Hfq. The RIL-seq library preparation protocol uses an rRNA depletion step to remove contaminating ribosomal RNA. For Hfq CLASH this is not necessary, and we show that chimeras containing rRNA fragments, which presumably represent noise, are not very abundant in our data (*Figure 2—figure supplements 1*, *2*, *4* and *5*). Our library preparation protocol also includes the use of random nucleotides in adapter sequences to remove potential PCR duplicates ('collapsing') from the data.

The very stringent purification conditions used in CLASH could, in some cases, also be a disadvantage as it completely relies on UV cross-linking to isolate directly bound RNAs. In cases where the efficiencies of protein-RNA cross-linking are low (for example, in the case of proteins that only recognize double-stranded RNA), RIL-seq may be a better approach as it does not completely rely on UV cross-linking (*Melamed et al., 2016*).

A large number of interactions were unique to both RIL-seq and Hfq CLASH datasets, which we believe can be explained by a number of technical and experimental factors. The denaturing purification conditions used with CLASH completely disrupt the Hfq hexamer (*Tree et al., 2014* and this work). Therefore, during the adapter ligation reactions the RNA ends are likely more accessible for ligation. In support of this, in the RIL-Seq data, the sRNAs are mostly found in the second half of the chimeras (*Melamed et al., 2016*), whilst in the Hfq CLASH data we observe sRNAs fragments with almost equal distributed in both sides (45% in left fragment and 55% in right fragment). Indeed, it was proposed that in RIL-seq the 3' end of the sRNA is buried in the hexamer and therefore not always accessible for ligation (*Melamed et al., 2016*).

For the RIL-seq experiments, the authors harvested the cells at 4°C and resuspended them in ice-cold PBS prior to UV irradiation (*Melamed et al., 2018*; *Melamed et al., 2016*). This procedure results in a cold-shock that can affect the sRNA-interactome as well as sRNA stability. We cross-link actively growing cells in their growth medium and we UV irradiate our cells within seconds using the Vari-X-linker we recently developed (*van Nues et al., 2017*). We use filtration devices to rapidly harvest our cells (less than 30 seconds) and the cells are subsequently stored on the filters at −80°C. We previously showed that filtration, combined with short UV cross-linking times dramatically reduces noise introduced by the activation of the DNA damage response and significantly increased the recovery of short-lived RNA species (*van Nues et al., 2017*). We speculate that many of the interactions that are unique to our Hfq CLASH data represent short-lived RNA duplexes that are preferentially captured with our UV cross-linking and rapid cell filtration setup.

## Biological significance of the interactions

One important question that needs to be addressed in the field is how many of the interactions that are recovered by high-throughput RNA-RNA interactome methodologies represent physiologically or biologically relevant base-pairing interactions. The analysis of the RIL-seq (*Melamed et al., 2016*) and our CLASH data showed that the predicted mRNA targets did not frequently show significant changes in gene expression upon over-expression of the sRNA. It is, of course, possible that sRNA base-pairing mostly affects mRNA translation and mRNA stability to a lesser extent. Hence,

approaches other than over-expression analyses may need to be included to verify the interaction networks. Ribosome profiling analyses on mutant strains should be helpful in determining whether the absence of the sRNA alters the association of mRNA targets with ribosomes (*Guo et al., 2014*; *Wang et al., 2015*), however, this is also a method not without challenges (*Mohammad et al., 2019*). Whilst this work was in progress, the Margalit group presented compelling evidence suggesting that many mRNA targets compete for Hfq and that the binding efficiency of Hfq to the targets primarily determines the regulatory outcome (*Faigenbaum-Romm et al., 2020*). Those mRNAs that were significantly affected by sRNA over-expression were also more frequently and reproducibly found in chimeras with the sRNA. This offers a plausible explanation for why we did not always observe enrichment of differentially expressed genes in putative mRNA targets recovered in a relatively low number of chimeras. Another aspect to consider is that over-expression of sRNAs will not only impact the direct targets. For example, over-expression of ArcZ in *Salmonella* revealed widespread changes in gene expression, presumably as a result of redistribution of Hfq over the transcriptome (*Papenfort et al., 2009*). As a result, a relatively small fraction of the differentially expressed genes will be represented in the CLASH/RIL-seq data, resulting in poor p-values.

One could argue that some of the interactions we present here may represent weak or stochastic interactions that do not have a biological function. For example, sRNAs can cycle on Hfq (reviewed in *Santiago-Frangos and Woodson, 2018*) and it is therefore conceivable that some of the sRNA-sRNA chimeras detected in our CLASH data happen to be two sRNAs that were in close proximity during their exchange on Hfq. Although it is not possible to quantify the number of such interactions, we would argue that they are not very abundant in our data. We purified Hfq and cross-linked RNAs under very stringent and completely denaturing conditions before we do the intermolecular ligation reactions. Because our purification conditions completely disrupt the Hfq hexamer (this work and *Tree et al., 2014*), transient interactions that do not involve (significant) base-pairing would only be detected if an Hfq *monomer* was UV cross-linked to both sRNAs simultaneously and if the available 5′ and 3′ ends are in close proximity. Considering the poor efficiency of UV cross-linking, the likelihood of this happening is very low. Secondly, we show that our chimeras, including those that are supported by only a few reads, have a high propensity to form stable duplexes in silico (*Figure 3*). Finally, for many sRNAs, we identified enriched sequence motifs in predicted mRNA targets that have significant sequence complementarity to sRNA seeds (*Figure 4*, *Figure 4—figure supplements 1–12*, *Figure 8—figure supplements 1–3*). Thus, we conclude that with the CLASH protocol weaker or stochastic interactions are not easily recovered. While the CLASH and the RIL-seq analyses agree that for some sRNAs the more frequent interactions are more likely to affect target mRNA stability, they also highlight that low-abundance interactions have strong complementarity and base-pairing potential, thus are genuine. The biological significance of these is yet to be determined, but one possibility is that many low-frequency interactions occur to confer robustness to the regulation of a few principal targets (*Jost et al., 2013*) and we speculate that these principal targets are condition-specific.

Surprisingly, for ArcZ and CyaR, even some of the mRNA targets found in a larger number of chimeric fragments were not differentially expressed. Possible explanations include their regulation at the protein synthesis level, but not at the RNA level, or control by fine-tuning, which would result in modest or undetectable changes in transcript levels.

## sRNA-sRNA interactions; ArcZ regulation of CyaR

One of the most striking observation of our global study was the abundance of sRNA-sRNA interactions *in E. coli*, many of which were growth-stage dependent. We experimentally validated the ArcZ-CyaR interaction, which involves the known seed sequence of ArcZ and the 5′end of CyaR. We demonstrate that ArcZ over-expression can reduce CyaR steady state levels but not *vice versa*, implying the regulation is unidirectional. Consistent with our findings, in *Salmonella*, over-expression of ArcZ showed a dramatic reduction in CyaR bound to Hfq and upregulation of CyaR targets, such as *nadE* (*Papenfort et al., 2009*). This suggests that this activity is conserved between these two Gram-negative bacteria. A similar type of unidirectional regulation has also been elegantly demonstrated for the Qrr3 sRNA of *Vibrio cholerae* (*Feng et al., 2015*). The fate of these sRNA-sRNA duplexes may depend on the position of the interaction; It was shown that if the interaction with Qrr3 involves its stabilizing 5′ stem-loop structure, the sRNA will be preferentially degraded (*Feng et al., 2015*). Consistent with this, folding of the chimeric reads suggests that ArcZ preferentially base-pairs with the 5′

end of CyaR (*Figure 6C* and *Figure 7A*). This may destabilize secondary structures that normally help to stabilize the sRNA.

The biological significance of ArcZ regulating CyaR remains unclear, however, a possible function could be to reduce noise in CyaR expression by preventing CyaR levels from overshooting during the transition phase. ArcZ and CyaR target mRNAs are associated with many different processes. Thus, these interactions are expected to connect multiple pathways. For example, ArcZ regulation of CyaR may connect adaptation to stationary phase/biofilm development (*De Lay and Gottesman, 2009*; *Monteiro et al., 2012*) to quorum sensing and cellular adherence (*De Lay and Gottesman, 2009*). CyaR expression is controlled by the global regulator Crp. Most of the genes controlled by Crp are involved in transport and/or catabolism of amino acids or sugar. Interestingly, ArcZ downre-gulates the *sdaCB* dicistron which encodes for proteins involved in serine uptake and metabolism (*Papenfort et al., 2009*). This operon has been shown to be regulated by Crp as well, suggesting that ArcZ can counteract the activity of Crp.

# Materials and methods

## Bacterial strains and culture conditions

An overview of the bacterial strains used in this study is provided in the Key Resources Table. The *E. coli* MG1655 (*Blattner et al., 1997*) and TOP10F' strains served as parental strains. The *E. coli* K12 strain used for CLASH experiments, MG1655 *hfq*::HTF was previously reported (*Tree et al., 2014*). Cells were grown in Lysogeny Broth (LB) at 37°C under aerobic conditions with shaking at 200 rpm. The media were supplemented with antibiotics where required at the following concentrations: chloramphenicol (Corning, –S, C239RI) - 25 µg/ml and kanamycin (Gibco, US,–11815–024) - 50 µg/ml. For induction of sRNA expression from plasmids, 1 mM IPTG, or 200 nM anhydrotetracycline hydrochloride (Sigma, 1035708–25 MG) were used.

## Construction of sRNA expression plasmids

The plasmids used in this study are listed in the Key Resources Table. The gene fragments and primers used for cloning procedures in this work are provided in *Supplementary file 10*. For the sRNA over-expression constructs, the sRNA gene of interest was cloned at the transcriptional +one site under P*lacO* control by amplifying the pZE12*luc* plasmid (Expressys) by inverse PCR using Q5 DNA Polymerase (NEB). The sRNA genes and seed mutants were synthesized as ultramers (IDT; *Supplementary file 10*) which served as the forward primers. The reverse primer (oligo pZE12_5-P_rev, *Supplementary file 10*) bears a monophosphorylated 5' end to allow blunt-end self-ligation. The PCR reaction was digested with 10U DpnI (NEB) for 1 hr at 37°C and purified by ethanol precipitation. The linear PCR product was circularized by self-ligation, and transformed in *E. coli* TOP10F' competent cells. Positive transformants were screened by Sanger sequencing (Edinburgh Genomics, Edinburgh, UK). Small RNA over-expression constructs derived from the pZA21MCS (Expressys) were generated identically, using the indicated ultramers in *Supplementary file 10* as forward primers and oligo pZA21MCS_5P_rev as the reverse primer.

## Hfq UV cross-linking, ligation and analysis of hybrids (Hfq-CLASH)

CLASH was performed essentially as described (*Waters et al., 2017*), with a number of modifications including changes in incubation steps, cDNA library preparation, reaction volumes and UV cross-linking. *E. coli* expressing the chromosomal Hfq-HTF were grown overnight in LB at 37°C with shaking (200 rpm), diluted to starter $OD_{600}$ 0.05 in fresh LB, and re-grown with shaking at 37°C in 750 ml LB. A volume of culture equivalent to 80 $OD_{600}$ per ml was removed at the following cell-densities ($OD_{600}$): 0.4, 0.8, 1.2, 1.8, 2.4, 3.0 and 4.0, and immediately subjected to UV (254 nm) irra-diation for 22 s (~500 mJ/cm2) in the Vari-X-linker (*van Nues et al., 2017*) (https://www.vari-x-link.com). Cells were harvested using a rapid filtration device (*van Nues et al., 2017*) (https://www.vari-x-link.com) onto 0.45 µM nitrocellulose filters (Sigma, UK, HAWP14250) and flash-frozen on the membrane in liquid nitrogen. Membranes were washed with ~15 ml ice-cold phosphate-buffered saline (PBS), and cells were harvested by centrifugation. Cell pellets were lysed by bead-beating in 1 vol per weight TN150 buffer (50 mM Tris pH 8.0, 150 mM NaCl, 0.1% NP-40, 5 mM β-mercaptoetha-nol) in the presence of protease inhibitors (Roche, A32965), and 3 volumes 0.1 mm Zirconia beads

(Thistle Scientific, 11079101z), by performing five cycles of 1 min vortexing followed by 1 min incubation on ice. One additional volume of TN150 buffer was added. To reduce the viscosity of the lysate and remove contaminating DNA, the lysate was incubated with RQ1 DNase I (10 U/ml Promega, M6101) for 30 min on ice. Two-additional volumes of TN150 were added and mixed with the lysates by vortexing. The lysates were centrifuged for 20 min at 4000 rpm at 4°C and subsequently clarified by a second centrifugation step at 13.4 krpm, for 20 min at 4°C. Purification of the UV cross-linked Hfq-HTF-RNA complexes and cDNA library preparation was performed as described (*Granneman et al., 2009*). Cell lysates were incubated with 50 µl of pre-equilibrated M2 anti-FLAG beads (Sigma, M8823-5ML) for 1–2 hr at 4°C. The anti-FLAG beads were washed three times 10 min with 2 ml TN1000 (50 mM Tris pH 7.5, 0.1% NP-40, 1M NaCl) and three times 10 min with TN150 without protease inhibitors (50 mM Tris pH 7.5, 0.1% NP-40, 150 mM NaCl). For TEV cleavage, the beads were resuspended in 250 µl of TN150 buffer (without protease inhibitors) and incubated with home-made GST-TEV protease at room temperature for 1.5 hr. The TEV eluates were then incubated with a fresh 1:100 dilution preparation of RNaceIt (RNase A and T1 mixture; Agilent, 400720) for exactly 5 min at 37°C, after which they were mixed with 0.4 g GuHCl (6M, Sigma, G3272-100G), NaCl (300 mM), and Imidazole (10 mM, I202-25G). Note this needs to be carefully optimized to obtain high-quality cDNA libraries. The samples were then transferred to 50 µl Nickel-NTA agarose beads (Qiagen, 30210), equilibrated with wash buffer 1 (6 M GuHCl, 0.1% NP-40, 300 mM NaCl, 50 mM Tris pH 7.8, 10 mM Imidazole, 5 mM beta-mercaptoethanol). Binding was performed at 4°C overnight with rotation. The following day, the beads were transferred to Pierce SnapCap spin columns (Thermo Fisher, 69725), washed three times with wash buffer 1 and three times with 1xPNK buffer (10 mM MgCl$_2$, 50 mM Tris pH 7.8, 0.1% NP-40, 5 mM beta-mercaptoethanol). The washes were followed by on-column TSAP incubation (Thermosensitive alkaline phosphatase, Promega, M9910) treatment for 1 hr at 37°C with 8 U of phosphatase in 60 µl of 1xPNK, in the presence of 80U RNasin (Promega, N2115). The beads were washed once with 500 µl wash buffer 1 and three times with 500 µl 1xPNK buffer. To add 3'-linkers (App-PE – Key Resources Table), the Nickel-NTA beads were incubated in 80 µl 3'-linker ligation mix with (1 X PNK buffer, 1 µM 3'-adapter, 10% PEG8000, 30U Truncated T4 RNA ligase 2 K227Q (NEB, M0351L), 60U RNasin). The samples were incubated for 4 hr at 25°C. The 5' ends of bound RNAs were radiolabelled with 30U T4 PNK (NEB, M0201L) and 3 µl $^{32}$P-γATP (1.1 µCi; Perkin Elmer, NEG502Z-500) in 1xPNK buffer for 40 min at 37°C, after which ATP (Roche, 11140965001) was added to a final concentration of 1 mM, and the incubation prolonged for another 20 min to complete 5' end phosphorylation. The resin was washed three times with 500 µl wash buffer one and three times with equal volume of 1xPNK buffer. For on-bead 5'-linker ligation, the beads were incubated 16 hr at 16°C in 1xPNK buffer with 40U T4 RNA ligase I (NEB, M0204L), and 1 µl 100 µM L5 adapter (Key Resources Table), in the presence of 1 mM AtP and 60U RNasin. The Nickel-NTA beads were washed three times with wash buffer one and three times with buffer 2 (50 mM Tris–HCl pH 7.8, 50 mM NaCl, 10 mM imidazole, 0.1% NP-40, 5 mM β-mercaptoethanol). The protein-RNA complexes were eluted in two steps in new tubes with 200 µl of elution buffer (wash buffer 2 with 250 mM imidazole). The protein-RNA complexes were precipitated on ice by adding TCA (T0699-100ML) to a final concentration of 20%, followed by a 20 min centrifugation at 4°C at 13.4 krpm. Pellets were washed with 800 µl acetone, and air dried for a few minutes in the hood. The protein pellet was resuspended and incubated at 65°C in 20 µl 1x NuPage loading buffer (Thermo Scientific, NP0007), resolved on 4–12% NuPAGE gels (Thermo Scientific, NP0323PK2) and visualised by autoradiography. The cross-linked proteins-RNA were cut directly from the gel and incubated with 160 µg of Proteinase K (Roche, 3115801001) in 600 µl wash buffer 2 supplemented with 1% SDS and 5 mM EDTA at 55°C for 2–3 hr with mixing. The RNA was subsequently extracted by phenol-chloroform extraction and ethanol precipitated. The RNA pellet was directly resuspended in RT buffer and was transcribed in a single reaction with the SuperScript IV system (Invitrogen, 18090010) according to manufacturer's instructions using the PE_reverse oligo as primer. The cDNA was purified with the DNA Clean and Concentrator 5 kit (Zymo Research) and eluted in 11 µl DEPC water. Half of the cDNA (5 µl) was amplified by PCR using Pfu Polymerase (Promega, M7745) with the cycling conditions (95°C for 2 min; 20–24 cycles: 95°C for 20 s, 52°C for 30 s and 72°C for 1 min; final extension of 72°C for 5 min). The PCR primers are listed in the Key Resources Table. PCR products were treated with 40U Exonuclease 1 (NEB, M0293L) for 1 hr at 37°C to remove free oligonucleotide and purified by ethanol precipitation/or the DNA Clean and Concentrator 5 kit (Zymo Research, D4003T). Libraries were resolved on a 2% MetaPhor agarose (Lonza,

LZ50181) gel and 175–300 bp fragments were gel-extracted with the MinElute kit (Qiagen, 28004) according to manufacturer's instructions. All libraries were quantified on a 2100 Bionalyzer using the High-Sensitivity DNA assay and a Qubit 4 (Thermo Scientific, Q33226). Individual libraries were pooled based on concentration and barcode sequence identity. Paired-end sequencing (75 bp) was performed by Edinburgh Genomics on an Illumina HiSeq 4000 platform.

## RNA-seq

*E. coli* MG1655 was cultured, UV-irradiated and harvested as described for the CLASH procedure. Total RNA was extracted using the Guanidium thiocyanate phenol method. RNA integrity was assessed with the Prokaryote Total RNA Nano assay on a 2100 Bioanalyzer (Agilent, G2939BA). Sequencing libraries from two biological replicates were prepared by NovoGene using the TruSeq library preparation protocol and 150 bp paired-end sequencing was performed on an Illumina Nova-Seq 6000 system. This yielded ~7–8 million paired-end reads per sample.

## Small RNA over-expression studies

Individual TOP10F' clones carrying pZA21 and pZE12-derived sRNA constructs and control plasmids combinations (Key Resources Table) were cultured to $OD_{600}$ 0.1 and expression of sRNAs was induced with IPTG and anhydrotetracycline hydrochloride (Sigma, I6758-1G and 1035708–25 MG) for 1 hr. Cells were collected by centrifugation for 30 s at 14,000 rpm, flash-frozen in liquid nitrogen and total RNA was isolated as above. Gene expression was quantified by RT-qPCR (see below) using 10 ng total RNA as template, and expressed as fold change relative to the reference sample containing pJV300 (*Sittka et al., 2007*) or empty pZA21.

## RT-qPCR

Total RNA (10 µg) was treated with 2 U of Turbo DNase (Thermo Scientific, AM2238) for 1 hr at 37°C in a 10 µl reaction in the presence of 2 U superaseIn RNase inhibitor (Thermo Scientific, AM2694). The RNA was purified with RNAClean XP beads (Beckman Coulter, A63987). Quantitative PCR was performed on 10 ng of DNAse I-treated total RNA using the Luna Universal One-Step RT-qPCR Kit (NEB, E3005E) according to manufacturer's instructions. The qPCRs were run on a LightCycler 480 (Roche), and the specificity of the product was assessed by generating melt curves, as follows: 65 °C-60s, 95°C (0.11 ramp rate with five acquisitions per °C, continuous). The data analyses were performed with the IDEAS2.0 software, at default settings: Absolute Quantification/Fit Points for Cp determination and Melt Curve Genotyping. The RT-qPCR for all samples was performed in technical triplicate. Outliers from the samples with technical triplicate standard deviations of Cp >0.3 were discarded from the analyses. To calculate the fold-change relative to the control, the $2^{-ddCp}$ method was employed, using 5S rRNA (*rrfD*) as the reference gene. Experiments were performed for three biological replicates, and the mean fold-change and standard error of the mean were computed. Unless otherwise stated, significance of the fold-change difference compared to the reference sample control (for which fold-change = 1 by definition) was tested with a one-sample t-test.

## Northern blot analysis

Total RNA was extracted from cell lysates by GTC-Phenol extraction. 10 µg total RNA was separated on an 8% polyacrylamide TBE-Urea gel and transferred to a nylon membrane (HyBond N+, GEHealthcare, RPN1210B) by electroblotting for 4 hr at 50 V. Membranes were pre-hybridised in 10 ml of UltraHyb Oligo Hyb (Thermo Scientific, AM8663) for 1 hr and probed with $^{32}$P-labeled DNA oligo at 42°C for 12–18 hr in a hybridization oven. The sequences of the probes used for Northern blot detection are detailed in *Supplementary file 10*. Membranes were washed twice with 2xSSC + 0.5% SDS solution for 10 min and visualized using a Phosphor imaging screen and FujiFilm FLA-5100 Scanner (IP-S mode). For detection of highly abundant species (5S rRNA) autoradiography was used for exposure.

## Western blot analyses

*E. coli* MG1655 Hfq::*htf* lysates using strains cultured, cross-linked, harvested and lysed in identical conditions as the CLASH experiments containing 40 µg protein were resolved on PAGE gels and transferred to a nitrocellulose membrane. The membranes were blocked for 1 hr in blocking solution

(5% non-fat milk in PBST (1X phosphate saline buffer, 0.1% Tween-20). To detect Hfq-HTF protein, the membrane was probed overnight at 4°C with the Rabbit anti-TAP polyclonal primary antibody (Thermo Fisher, 1:5000 dilution in blocking solution), which recognizes an epitope at the region between the TEV-cleavage site and His6. For the loading control we used a rabbit polyclonal to GroEL primary antibody (Abcam, 1:150000 dilution, ab82592), for 2 hr at room temperature. After 3 × 10 min PBST washes, the membranes were blotted for one hour with a Goat anti-rabbit IgG H and L (IRDye 800) secondary antibody (Abcam, ab216773, 1:10,000 in blocking solution) at room temperature. Finally, after three 10 min PBST washes, the blot was rinsed in PBS, and the proteins were visualised with a LI-COR (Odyssey CLx) using the 800 nm channel and scan intensity 4. Image acquisition and quantifications were performed with the Image Studio Software.

## Computational analysis

### Pre-processing of the raw sequencing data

Raw sequencing reads in fastq files were processed using a pipeline developed by Sander Granneman, which uses tools from the pyCRAC package (*Webb et al., 2014*). The entire pipeline is available at https://bitbucket.org/sgrann/). The CRAC_pipeline_PE.py pipeline first demultiplexes the data using pyBarcodeFilter.py and the in-read barcode sequences found in the L5 5' adapters. Flexbar then trims the reads to remove 3'-adapter sequences and poor-quality nucleotides (Phred score <23). Using the random nucleotide information present in the L5 5' adaptor sequences, the reads are then collapsed to remove potential PCR duplicates. The reads were then mapped to the *E. coli* MG1655 genome using Novoalign (www.novocraft.com). To determine to which genes the reads mapped to, we generated an annotation file in the Gene Transfer Format (GTF). This file contains the start and end positions of each gene on the chromosome as well as what genomic features (i.e. sRNA, protein- coding, tRNA) it belongs to. To generate this file, we used the Rockhopper software (*Tjaden, 2015*) on *E. coli* rRNA-depleted total RNA-seq data (generated by Christel Sirocchi), a minimal GTF file obtained from ENSEMBL (without UTR information). The resulting GTF file contained information not only on the coding sequences, but also complete 5' and 3' UTR coordinates. We then used pyReadCounters.py with Novoalign output files as input and the GTF annotation file to count the total number of unique cDNAs that mapped to each gene.

### Normalization steps

To normalize the read count data generated with pyReadCounters.py and to correct for differences in library depth between time-points, we calculated Transcripts Per Million reads (TPM) for each gene. Briefly, for each time-point the raw counts for each gene was first divided by the gene length and then divided by the sum of all the values for the genes in that time-point to normalize for differences in library depth. The TPM values for each $OD_{600}$ studied were then $log_2$-normalized.

### Hfq-binding coverage plots

For the analysis of the Hfq binding sites the pyCRAC package (*Webb et al., 2014*) was used (versions. 1.3.2–1.4.3). The pyBinCollector tool was used to generate Hfq cross-linking distribution plots over genomic features. First, PyCalculateFDRs.py was used to identify the significantly enriched Hfq-binding peaks (minimum 10 reads, minimum 20 nucleotide intervals). Next, pyBinCollector was used to normalize gene lengths by dividing their sequences into 100 bins and calculate nucleotide densities for each bin. To generate the distribution profile for all genes individually, we normalized the total number of read clusters (assemblies of overlapping cDNA sequences) covering each nucleotide position by the total number of clusters that cover the gene. Motif searches were performed with pyMotif.py using the significantly enriched Hfq-binding peaks (FDR intervals). The 4–8 nucleotide k-mers with Z-scores above the indicated threshold were used for making the motif logo with the k-mer probability logo tool (*Wu and Bartel, 2017*) with the -ranked option (http://kplogo.wi.mit.edu/).

### Analysis of chimeric reads

Chimeric reads were identified using the hyb package using default settings (*Travis et al., 2014*) and further analysed using the pyCRAC package (*Webb et al., 2014*). To apply this single-end specific pipeline to our paired-end sequencing data, we joined forward and reverse reads using FLASH

(https://github.com/dstreett/FLASH2) (*Magoč and Salzberg, 2011*), which merges overlapping paired reads into a single read. Paired reads that were not considered overlapping were subsequently concatenated into a single sequence and again filtered for overlapping reads that were missed by FLASH. These were then analysed using hyb. The -anti option for the hyb pipeline was used to be able to use a genomic *E. coli* hyb database, rather than a transcript database. Uniquely annotated hybrids (.ua.hyb) were used in subsequent analyses. To visualise the hybrids in the genome browser, the. ua.hyb output files were converted to the GTF format. To generate distribution plots for the genes to which the chimeric reads mapped, the parts of the chimeras were clustered with pyClusterReads.py and BEDtools (*Quinlan and Hall, 2010*) (intersectBed) was used to remove clusters that map to multiple regions. To produce the coverage plots with pyBinCollector, each cluster was counted only once, and the number of reads belonging to each cluster was ignored.

## Statistical filtering of the data

The uniquely annotated chimeras from the merged CLASH experiments were statistically scored using available pipelines (*Waters et al., 2017*). Only chimeras with an Benjamini-Hochberg adjusted p-value lower than 0.05 were considered and referred to as statistically filtered chimeras.

## Predicted folding energy analyses

Cumulative distributions of minimum folding energy were generated using the minimum folding energies predicted with RNADuplex (*Lorenz et al., 2011*) for all statistically filtered sRNA-mRNA chimeras. To generate the data for the shuffled chimeras, the fragments were randomly shuffled over the same gene, or over genes belonging to the same class of genes (e.g sRNAs or mRNAs), respectively. Significance was tested with the Kolmgorov-Smirnov test.

## Motif analyses for sRNA targets

For each sRNA with at least five different putative targets, we clustered those chimeras based on the similarity of sRNA sequences using K-means clustering. The clustering step was skipped for those sRNAs for which almost all chimeric reads overlapped the same region. The sequences of the fused mRNA fragments in each cluster were extracted and motif searches using MEME (*Bailey et al., 2009*). To calculate complementarity between the identified motifs in putative mRNA targets and the sRNA we used MAST (*Bailey et al., 2009*). Only motifs that had a MAST p-value<=0.001 were considered.

## Microarray analyses

ArcZ, Spot42 and GcvB microarray data were processed by GEO2R using the limma package (*Ritchie et al., 2015*). The accession numbers for these datasets are GSE17771, GSE24875 and GSE26573. The processed CyaR data were obtained from the Supplementary data provided in the paper describing the CyaR over-expression in *E. coli* (*De Lay and Gottesman, 2009*). Cumulative distribution plots were generated using the T-statistics calculated by the limma package. Average expression levels were calculated by averaging the expression of genes in the parental and over-expression strain.

## sRNA density plots

To visualize the nucleotide read density of sRNA-target pairs for a given sRNA, the hit counts at each nucleotide position for all statistically filtered chimeras were summed. The count data was $log_2$-normalized (actually $log_2$(Chimera count +1) to avoid NaN for nucleotide positions with 0 hits when log-transforming the data).

To make distributions of the chimeric reads around known sRNA and mRNA seeds, we manually retrieved the experimentally validated sRNA and mRNA seed sequences from sRNATarbase 3.0 (*Wang et al., 2015*) and literature. We converted the FASTA sequences to the genomic coordinates of our reference genome. Next, we normalized the length of all sequences to eight nucleotides with pyNormalizeIntervalLengths.py, then used the pyBinCollector tool to calculate the overlap of the intervals corresponding to statistically filtered chimeric reads with the seed sequence interval of each sRNA and sRNA-mRNA interaction. sRNA-sRNA network visualization.

Only the sRNA-sRNA chimeric reads representing statistically filtered chimeras in the merged CLASH dataset were considered. For each such interaction, chimera counts corresponding in either orientation were summed, $\log_2$-transformed and visualized with the igraph Python package.

## Data and code availability

The next generation sequencing data have been deposited on the NCBI Gene Expression Omnibus (GEO) with accession number GSE123050. The python pyCRAC (*Webb et al., 2014*), kinetic-CRAC and GenomeBrowser software packages used for analysing the data are available from https://bit-bucket.org/sgrann (pyCRAC up to version 1.4.3), https://git.ecdf.ed.ac.uk/sgrannem/ and pypi (https://pypi.org/user/g_ronimo/). The hyb pipeline for identifying chimeric reads is available from https://github.com/gkudla/hyb. The scripts for statistical analysis of hyb data is available from https://bitbucket.org/jaitree/hyb_stats/. The FLASH algorithm for merging paired reads is available from https://github.com/dstreett/FLASH2. Bedgraph and Gene Transfer Format (GTF) generated from the analysis of the Hfq CLASH, RNA-seq and TEX RNA-seq data (*Thomason et al., 2015*) are available from the Granneman lab DataShare repository (https://datashare.is.ed.ac.uk/handle/10283/2915).

## Acknowledgements

We are grateful to Lionello Bossi and Meriem El Karoui for their valuable feedback on the project and fruitful discussions. We thank Jörg Vogel and Yanjie Chao for providing the *Salmonella* CpxQ microarray data, Alasdair Ivens for help with the microarray data analysis, Christel Sirocchi for help with preparing *E. coli* RNA-seq libraries, Erica de Leau for expert technical assistance and the members of the Granneman lab for critically reading the manuscript. This work was supported by grants from the Wellcome Trust (091549 to SG and 102334 to IAI), the Wellcome Trust Centre for Cell Biology core grant (092076), a Medical Research Council non Clinical Senior Research Fellowship (MR/R008205/1 to SG), the Australian National Health and Medical Research Council Project grants (GNT1067241 and GNT1139313 to JJT) and the Autonomous Province of Trento (Axonomix to GV and MM). Next Generation Sequencing was in part carried out by Edinburgh Genomics that is supported through core grants from NERC (R8/H10/56), MRC (MR/K001744/1) and BBSRC (BB/J004243/1).

## Additional information

### Funding

| Funder | Grant reference number | Author |
| --- | --- | --- |
| Wellcome | 102334 | Ira Alexandra Iosub |
| Wellcome | 091549 | Sander Granneman |
| Medical Research Council | MR/R008205/1 | Sander Granneman |
| National Health and Medical Research Council | GNT1067241 | Jai J Tree |
| National Health and Medical Research Council | GNT1139313 | Jai J Tree |
| Axonomix | | Gabriella Viero<br>Marta Marchioretto |

The funders had no role in study design, data collection and interpretation, or the decision to submit the work for publication.

### Author contributions

Ira Alexandra Iosub, Conceptualization, Resources, Data curation, Software, Formal analysis, Supervision, Validation, Investigation, Visualization, Methodology, Writing - original draft, Project administration, Writing - review and editing; Robert Willem van Nues, Conceptualization, Resources, Supervision, Writing - review and editing; Stuart William McKellar, Resources, Methodology; Karen

Jule Nieken, Formal analysis, Validation; Marta Marchioretto, Brandon Sy, Investigation, Methodology; Jai Justin Tree, Resources, Software, Methodology, Writing - review and editing; Gabriella Viero, Resources, Formal analysis, Investigation, Methodology, Writing - review and editing; Sander Granneman, Conceptualization, Resources, Data curation, Software, Formal analysis, Supervision, Funding acquisition, Validation, Investigation, Visualization, Methodology, Writing - original draft, Project administration, Writing - review and editing

**Author ORCIDs**
Ira Alexandra Iosub (iD) https://orcid.org/0000-0002-2924-2471
Stuart William McKellar (iD) http://orcid.org/0000-0003-0792-9878
Gabriella Viero (iD) http://orcid.org/0000-0002-6755-285X
Sander Granneman (iD) https://orcid.org/0000-0003-4387-1271

**Decision letter and Author response**
Decision letter https://doi.org/10.7554/eLife.54655.sa1
Author response https://doi.org/10.7554/eLife.54655.sa2

## Additional files

**Supplementary files**
• Supplementary file 1. Hyb pipeline output from the merged Hfq CLASH data. Chromosome indicates the *E. coli* chromosome, sequence start and sequence end are the positions in the chimeric read that correspond to the first and second fragment. Chromosome start and chromosome end are the positions in the *E. coli* K12 reference genome.

• Supplementary file 2. Statistically filtered data. Chimeric reads were subsequently analyzed using a statistical pipeline described by *Waters et al., 2017*. Only chimeric reads that had a Benjami-Hochberg adjusted p-value (bh_adj_p_value) of 0.05 or less were considered The last three columns indicate in which growth phases the interactions were identified. Min. MFE indicates the minimal folding energies of the chimera, which was calculated using RNADuplex from the ViennaRNA package (*Lorenz et al., 2011*). The two pairs in the intermolecular base-pairs and structure columns are separated by ' and ".

• Supplementary file 3. Overview of sRNA-mRNA interactions found in the Hfq CLASH data and compared to the RIL-seq data. Shown are the statisitcally filtered sRNA-mRNA interactions identified in the Hfq CLASH data. Genomic sequences of the sRNA and mRNA fragments found in the chimeras are also provided. Total_hybrids indicates the total number of interactions involving these sequences that were found. Min. MFE indicates the minimal folding enrgies of the chimera, which was calculated using RNADuplex from the ViennaRNA package (*Lorenz et al., 2011*). The last column indicates which of the sRNA-mRNA interactions were also found in the RIL-seq S-chimera data (*Melamed et al., 2016*).

• Supplementary file 4. Overview of sRNA-sRNA interactions found in the Hfq CLASH data and compared to the RIL-seq data. Shown are the statistically filtered sRNA-sRNA interactions identified in the Hfq CLASH data. Genomic sequences of the sRNA fragments found in the chimeras are also provided. Total_hybrids indicates the total number of interactions involving these sequences that were found. Min. MFE indicates the minimal folding enrgies of the chimera, which was calculated using RNADuplex from the ViennaRNA package (*Lorenz et al., 2011*). The last column indicates which of the sRNA-mRNA interactions were also found in the RIL-seq S-chimera data (*Melamed et al., 2016*).

• Supplementary file 5. Overview of putative 3'UTR derived sRNAs. 3'UTR-mRNA and mRNA-3'UTR interactions were isolated from the statistically filtered data and compared against the RILseq data (*Melamed et al., 2016*), Salmonella TIERseq data (*Chao et al., 2012*) and RNA-seq data that was used transcription start sites in *E. coli* (*Thomason et al., 2015*). TEX insensitive are RNA fragments in 3'UTRs that are not sensitive to Terminator 5'-Phosphate Dependent Exonuclease treatment and therefore may be generated by an independent promoter. TEX sensitive are RNA fragments that likely have 5' monophosphates as, according to the TEX data, they were degraded by TEX.

• Supplementary file 6. Overview of 3'UTR-mRNA interactions found in the Hfq CLASH data and compared to the RIL-seq data. Shown are the statistically filtered 3'UTR-mRNA interactions identified in the Hfq CLASH data. Genomic sequences of the 3'UTR and mRNA fragments found in the chimeras are also provided. Total_hybrids indicates the total number of interactions involving these sequences that were found. Min. MFE indicates the minimal folding enrgies of the chimera, which was calculated using RNADuplex from the ViennaRNA package (*Lorenz et al., 2011*). The last column indicates which of the sRNA-mRNA interactions were also found in the RIL-seq S-chimera data (*Melamed et al., 2016*). The mRNA fragment location column indicates where in the mRNA target the putative 3'UTR-derived sRNA was base-paired.

• Supplementary file 7. Experimentally validated interactions in the statistically filtered Hfq CLASH data. Chimeric reads were analyzed using a statistical pipeline described by *Waters et al., 2017*. Only chimeric reads that had a Benjami-Hochberg adjusted p-value (bh_adj_p_value) of 0.05 or less were considered. Shown are the sRNA-mRNA interactions that were experimentally validated, retrieved from sRNATarbase 3.0 (*Wang et al., 2016*) and recent literature (*Bianco et al., 2019*; *Chao and Vogel, 2016*; *De Mets et al., 2019*; *Guo et al., 2014*; *Lalaouna et al., 2015b*; *Miyakoshi et al., 2018*). Min. MFE indicates the minimal folding energies of the chimera, which was calculated using RNADuplex from the ViennaRNA package (*Lorenz et al., 2011*). The last three columns indicate in which growth phases the interactions were identified.

• Supplementary file 8. Motif analyses of chimeric fragments that mapped to 5' UTRs. PyMotif from the pyCRAC package was used for these analyses. For the motif search analyses we first clustered overlapping chimeric fragments into a single contig. For the 5'UTR motif analyses we used 356 clusters.

• Supplementary file 9. Motif analyses of chimeric fragments that mapped to 3' UTRs. PyMotif from the pyCRAC package was used for these analyses. For the motif search analyses we first clustered overlapping chimeric fragments into a single contig. For the 3'UTR motif analyses we used 188 clusters.

• Supplementary file 10. Oligonucleotides used in this study.

• Supplementary file 11. Key Resources Table.

• Transparent reporting form

## Data availability

The next generation sequencing data have been deposited on the NCBI Gene Expression Omnibus (GEO) with accession number GSE123050. The python pyCRAC (Webb et al., 2014), kinetic-CRAC and GenomeBrowser software packages used for analysing the data are available from https://bit-bucket.org/sgrann (pyCRAC up to version 1.4.3), https://git.ecdf.ed.ac.uk/sgrannem/ and pypi (https://pypi.org/user/g_ronimo/). The hyb pipeline for identifying chimeric reads is available from https://github.com/gkudla/hyb. The scripts for statistical analysis of hyb data is available from https://bitbucket.org/jaitree/hyb_stats/. The FLASH algorithm for merging paired reads is available from https://github.com/dstreett/FLASH2. Bedgraph and Gene Transfer Format (GTF) generated from the analysis of the Hfq CLASH, RNA-seq and TEX RNA-seq data (Thomason et al., 2015) are available from the Granneman lab DataShare repository (https://datashare.is.ed.ac.uk/handle/10283/2915).

The following datasets were generated:

| Author(s) | Year | Dataset title | Dataset URL | Database and Identifier |
|---|---|---|---|---|
| Granneman S | 2020 | Hfq CLASH uncovers sRNA-target interaction networks involved in adaptation to nutrient availability | https://www.ncbi.nlm.nih.gov/geo/query/acc.cgi?acc=GSE123050 | NCBI Gene Expression Omnibus, GSE123050 |
| Granneman S | 2019 | Hfq CLASH and TEX processed data | https://doi.org/10.7488/ds/2537 | Edinburgh DataShare, 10.7488/ds/2537 |

The following previously published datasets were used:

| Author(s) | Year | Dataset title | Dataset URL | Database and Identifier |
|---|---|---|---|---|
| Melamed S, Peer A, Faigenbaum-Romm R, Gatt YE, Reiss N, Bar A, Altuvia Y, Argaman L, Margalit H | 2016 | Transcriptome wide mapping of Hfq mediated RNA-RNA interactions in E. coli (RIL-seq) | https://www.ebi.ac.uk/arrayexpress/experiments/E-MTAB-3910/ | ArrayExpress, E-MTAB-3910 |
| Faigenbaum-Romm R, Reich A, Gatt YE, Barsheshet M, Argaman L, Margalit H | 2020 | RNA-seq data of Escheirchia coli K12 MG1655 of overexpression of GcvB, MicA, ArcZ, RyhB or CyaR sRNAs | https://www.ebi.ac.uk/arrayexpress/experiments/E-MTAB-8229/ | ArrayExpress, E-MTAB-8229 |
| Lalaouna D, Eyraud A, Devinck A, Prévost K, Massé E | 2018 | MS2-affinity purification coupled with RNA sequencing (MAPS) reveals GcvB sRNA targetome. | https://www.ncbi.nlm.nih.gov/geo/query/acc.cgi?acc=GSE80019 | NCBI Gene Expression Omnibus, GSE80019 |
| Lalaouna D, Prévost K, Laliberté G, Houé V, Massé E | 2018 | MS2-affinity purification coupled with RNA sequencing (MAPS) reveals CyaR sRNA targetome in Escherichia coli. | https://www.ncbi.nlm.nih.gov/geo/query/acc.cgi?acc=GSE90128 | NCBI Gene Expression Omnibus, GSE90128 |
| Papenfort K, Said N, Welsink T, Lucchini S, Hinton JC, Vogel J | 2009 | Specific and pleiotropic patterns of mRNA regulation by ArcZ | https://www.ncbi.nlm.nih.gov/geo/query/acc.cgi?acc=GSE17771 | NCBI Gene Expression Omnibus, GSE17771 |
| Beisel CL, Storz G | 2011 | The base pairing RNA Spot 42 participates in a multi-output feedforward loop to help enact catabolite repression in Escherichia coli | https://www.ncbi.nlm.nih.gov/geo/query/acc.cgi?acc=GSE24875 | NCBI Gene Expression Omnibus, GSE24875 |
| Sharma CM, Papenfort K, Pernitzsch SR, Mollenkopf H, Hinton JC, Vogel J | 2011 | Global post-transcriptional control of genes involved in amino acid metabolism by the Hfq-dependent GcvB RNA | https://www.ncbi.nlm.nih.gov/geo/query/acc.cgi?acc=GSE26573 | NCBI Gene Expression Omnibus, GSE26573 |

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
