## [Decision Letter]

**Acceptance summary:**

This study uses a genome-scale approach, CLASH, to identify many RNA-RNA interactions in *Escherichia coli*. The interacting RNA pairs identified in this work represent a valuable resource for groups studying RNA-based regulation in bacteria. Moreover, the data reveal many interacting pairs of small, regulatory RNAs (sRNAs), suggesting complex regulatory cross-talk among sRNAs.

**Decision letter after peer review:**

[Editors’ note: the authors submitted for reconsideration following the decision after peer review. What follows is the decision letter after the first round of review.]

Thank you for submitting your work entitled "Hfq CLASH uncovers sRNA-target interaction networks involved in adaptation to nutrient availability" for consideration by *eLife*. Your article has been reviewed by two peer reviewers, and the evaluation has been overseen by a Reviewing Editor and a Senior Editor. The following individual involved in review of your submission has agreed to reveal their identity: Ben F Luisi (Reviewer #2).

Our decision has been reached after consultation between the reviewers and the Reviewing Editor. Based on these discussions and the individual reviews below, we regret to inform you that your work cannot be considered further for publication in *eLife*.

While the reviewers are enthusiastic about the potential of the resource that the CLASH data represent, concerns were raised about the validation of these data. Additionally, the reviewers felt that the follow-up studies are interesting, but that some of the conclusions need to be softened. With additional validation of the CLASH data, the manuscript would likely be suitable for publication in *eLife*, without the need for much in the way of new experimental data. Nonetheless, the required analyses will likely take some time. We encourage you to resubmit if you can make a more compelling case that the CLASH data represent physiological RNA-RNA interactions.

The major concern is that there is currently insufficient evidence to conclude that the RNA-RNA pairs identified by CLASH represent bona fide RNA-RNA pairs inside cells. Many of the reported RNA-RNA pairs appear to have been identified only once, and many include highly abundant RNAs (i.e. tRNA, rRNA). Moreover, overlap with RIL-seq data is fairly limited. While the discussion clearly lays out why overlap with RIL-seq data might be low, this also raises the bar for validating the RNA-RNA pairs not found by RIL-seq. It should be possible to use bioinformatic analyses to further test whether the novel RNA-RNA pairs are genuine. For example, are novel sRNA targets enriched for sequences complementary to the sRNA seeds? Are known sRNA-regulated genes enriched for sRNA-mRNA pairs, and vice versa? These analyses are most important for the novel RNA-RNA pairs identified by CLASH (i.e. not found by RIL-seq).

Reviewer #1:

Overall, this is very interesting work, and the manuscript obviously represents a great deal of effort. My major criticisms of the work are two-fold. First, the manuscript seems very diffuse – touching on too many topics at a rather surface level. Second, the biological implications of the experimental results are overstated. I hope my comments will be useful to the authors as they consider how to revise their manuscript.

1) The title of the manuscript is misleading. The main functional characterization of MdoR and its targets is intriguing and hints at a physiological function related to carbon source adaptation, but there is a long way to go to say that this is truly the function of this sRNA.

2) ArcZ-CyaR experiment in the middle is not well connected to the rest of the manuscript. The inclusion in the model figure doesn't really help shed light on the biological role for this interaction.

3) Subsection “Hfq CLASH predicts sRNA-sRNA interactions as a widespread layer of post transcriptional regulation”, third paragraph. Figure 5D, the wild-type ArcZ still affects mutant CyaR levels. The authors provide a hand-waving explanation that could be tested. Moreover, the authors state that ArcZ promotes CyaR degradation, but there is no direct evidence for this. It could be tested. Not sure it's the highest priority for this manuscript, given that this experiment in general is not well integrated. But at least the authors should modulate their statement to reflect the actual data.

4) Abstract – there is no direct evidence that MdoR enhances maltose uptake.

5) I did not understand the logic behind the analyses in Figure 2. The authors state that it was "logical to assume that changes in Hfq binding would also be reflected in changes in sRNA steady state levels." However, there are numerous studies showing that different sRNAs bind Hfq via different modes, and that there is a great deal of variability regarding the role of Hfq in stabilizing sRNAs. Moreover, the competition among RNAs for binding to a limiting pool of Hfq will certainly change over time, and be influenced by the total sRNA abundance and any given sRNA's proportion of the total RNA pool. There seems to be no overall conclusion from the figure, and no follow up, so I would recommend deleting it.

6) Subsection “MdoR directly regulates the expression of major outer membrane porins and represses the envelope stress response pathway”, fourth paragraph: The authors state hypotheses in this section that are not further tested, and are not supported by data shown. These are more appropriate for modest speculation in the Discussion.

7) Figure 7F: Is the effect of MdoR SM on MicA significant?

8) The only MdoR-target interaction that was definitively demonstrated was MdoR-ompC, and indeed, the authors went above and beyond with evidence here. It is interesting that ompC levels are reduced in maltose (Figure 8B), but this is clearly NOT MdoR-dependent (Figure 8D). The differences in MicA and lamB RNA levels in the mdoR mutant grown in maltose are intriguing, but these effects can't be linked to a specific MdoR-target regulation. Minimally, the authors should try to make the link between molecular interaction of MdoR and a target (*rpoE*?) and the differences in MicA/lamB more clear.

9) Subsection “MdoR enhances maltoporin expression during maltose fermentation”, last paragraph: It would be very exciting if the data directly supported this statement. However, the experiments presented fall short. More physiological evidence is needed – growth phenotypes, maltose uptake assays, etc. In the absence of these, the authors must tone down their claims.

10) Because there is so little investigation of the physiology, the discussion of the physiological relevance of these findings is very superficial. The transition from exponential to stationary phase growth has been studied in *E. coli* growing in LB. What becomes limiting? The authors say very generically "the most favorable nutrients" become limiting. The finding that malEFG and MdoR are specifically expressed during a very narrow window of time in LB grown cells is very interesting. There must be more to the story of their regulation than malT-dependent maltose-inducible expression given this expression pattern in LB given that the main carbon source in LB is peptides/amino acids. The authors should work on improving the quality of the discussion of these issues, and be up front about the limitations of their study in this regard.

11) One key issue that should be addressed in the Discussion is the fact that these global approaches have so little overlap. I did appreciate the thorough description of the relative advantages provided by the Hfq-CLASH method as compared to RIL-seq. However, I think the field as a whole needs to find a way to discern direct, physiologically-relevant interactions from those that may be transient, weaker, and stochastic. I don't expect the authors to solve this issue, but it should be acknowledged. The sensitivity and accuracy of various methods needs a thorough investigation. At least, the authors could consider their Hfq-CLASH results in light of their total expression profiles (RNA-seq) of well characterized sRNAs and their regulons. What's the false negative rate for known interactions?

Reviewer #2:

This manuscript analyses the RNAs associated with the RNA chaperone Hfq in Escherchia coli at different growth stages, and in particular during the transition between stages. There has been other work published in this topic, but the new aspect of the work presented here is the depth of analysis of the transitions and the in depth characterisation of the associated RNAs. One important finding from this study is that sRNA expression does not correlate strongly with Hfq binding profile – suggesting that there must be context dependent binding of the RNA to Hfq. Another is the model for the regulatory network involving the processed transcript from the *mal* operon. The experimental work is extensive and there are many interesting new findings reported. There are several comments listed below that will hopefully be useful for the authors to consider:

1) Hfq for CLASH has two large tags on C-terminus. As the C-terminus has been proposed to participate in RNA/protein partners binding and Hfq autoinhibition (work from the Woodson group), have the authors done any controls to make sure this does not interfere with RNA banding/introduce false results?

2) "Hfq binds to sRNA-target RNA duplexes" – are RNA duplexes the only Hfq targets? For example, sRNAs were shown to cycle on Hfq, therefore one can imagine a situation in which one sRNA is not fully displaced and the second one already bound. Could some of the sRNA hybrids represent such state?

3) 'tRNA-tRNA and rRNA-rRNA chimeras originating from different coding regions were removed' why?

4) Can the authors please comment on other chimeras isolated, others than sRNA-mRNA and mRNA-mRNA? Would these represent Hfq targets in the cell?

5) Figure 3 – it is not clear what the enriched motifs are showing, 5' end of the chimera? 5' end of both RNAs in the chimera? Only mRNAs?

6) Explain in more detail what is meant by scrambled RNA.

7) Figure 4 – as only mutations in ArcZ cause disruption of the regulation, can it be an indirect effect, not the result of direct sRNA-sRNA regulation?

8) Figure 5B – It is difficult to see expression of ygaM (or YgaN, which the blot may be showing). Where is MdoR on the blot? If it is labelled malG it is somewhat confusing, as the blot presumably shows the sRNA fragments, not the whole mRNAs?

9) The signal for RyhB on Figure 5B is quite strong for OD 1.2 and 1.8, however on Figure 6C it is very weak. Can the authors explain? MdoR intensities seem to match, so presumably the RNAs quantities used are similar?

10) Is there an evidence that RyhB is in the cell as 5'PPP RNA? Perhaps it is not processed, but has the possibility of it harbouring a different 5' end has been excluded?

11) Figure 7C – It is confusing that the authors label 5' and 3' ends which are not real ends, and are different for each panel for MdoR. Could they mark, e.g. with dots, that these are not real ends of RNAs? Or indicate positions of the nucleotides shown? It would make analysing the results much easier.

12) Figure 7D- If an empty plasmid is used as a control and the blot probed for MdoR, can the authors explain what is being expressed in their control after 20 minutes? There may be a typo in the legend as it states that the samples were harvested 15 minutes after induction, but the blot shows the results for 20. What is the meaning of the red rectangle over the 15 minutes into MdoR expression?

13) Figure 7F – explain MdoR SM, it also isn't introduced in the text. Why does the seed mutation cause higher target levels? For RyeA it doesn't seem like the seed mutation has abolished regulation. Is RybB regulating MicA as well?

14) Have the authors tested how their substantial MdoR seed mutation influences RNA structure? Is it possible that, as the seed seems internal, the overall structure of the sRNA is disrupted and therefore the regulation lost? Can the mutant still bind to Hfq? The structure change would also explain problems with RNase E processing.

15) 'Notably, the fully-processed mutant MdoR sRNA is less abundant than the wild-type (Figure 9C) and longer (unprocessed) fragments that contain upstream malG regions could be readily detected (Figure 9E) '- should be 8C and 8E

16) 'We conclude that the dynamics of sRNA expression and binding to Hfq are not always highly correlated.' Any thoughts why?

17) Polysome preps used cyclohexamide, but this acts but blocking the peptide exit channel in the ribosome and may not trap polysomes except by blocking the last ribosome on the assembly. Another antibiotic or non-hydrolysable GTP might be better.

18) These references have related information that may be useful to comment on in the manuscript: de Mets, van Melderen and Gottesman, 2018; Miyakoshi et al., 2018.

Also, Hfq has been known to be involved in nutrient uptake regulation in *Pseudomonas* aeruginosa, where it inhibits translation of certain mRNAs depending on which nutrients are available. Pei et al., 2019, have solved high resolution structures of Hfq in complex with a target mRNA and other effector molecules to show how this Hfq based regulatory complex works. This research may be related to the theme of the report here and it might be helpful to comment on these findings.

[Editors’ note: further revisions were suggested prior to acceptance, as described below.]

Thank you for resubmitting your work entitled "Hfq CLASH uncovers sRNA-target interaction networks linked to nutrient availability adaptation" for further consideration by *eLife*. Your revised article has been reviewed by three peer reviewers, one of whom is a member of our Board of Reviewing Editors, and the evaluation has been overseen by James Manley as the Senior Editor.

All the reviewers were enthusiastic about the manuscript and recommend acceptance pending some edits to the text. In particular, the reviewers felt that the new analyses of the CLASH data make a strong case that the identified RNA-RNA interactions are real, and thus greatly expand the known set of interactions for *E. coli*, and reveal important insights such as the abundance of sRNA-sRNA interactions. To better focus the manuscript, we recommend removing the section on MdoR. While the reviewers found this work to be of interest, they also felt that it was peripheral to the main theme of the study, and would be better suited to an independent publication in a more specialized journal. This would free up some space in the paper to move some of the supplementary figure panels into the main figures, improving readability. Reviewer 3 has some specific suggestions for supplementary figure panels that could be moved into the main set of figures. The detailed reviews are listed below:

Reviewer #1:

The authors have provided further experimental data and analysis and made compelling response to most of the points raised in the review. The manuscript has been improved and the support for the conclusions strengthened considerably.

One minor issue is the Figure 2E legend does not explain the figure very clearly.

Reviewer #2:

This is a much improved revised version of a manuscript describing a global method for characterization of RNA-RNA interactions. The authors have nicely addressed my previous concerns and I have no additional major issues.

Reviewer #3 – :

The new analyses of the CLASH data make a very convincing case that the novel RNA-RNA pairs reflect real in vivo interactions. My preference would be to remove the MdoR story, which is interesting but peripheral to the main theme of the paper, and does not look at a novel sRNA (MdoR was identified previously by RIL-seq). Moreover, I suggest moving some of the more important supplementary figure panels into the main part of the paper.

Figure 2—figure supplement 1C. The "distance from sRNA seed" numbers appear to be similar to the length of the sRNAs. The authors should indicate the sRNA lengths.

Figure 2—figure supplement 6 (predicted base-pairing strength for identified interactions). This is an important analysis and should be moved to the main figures.

Figure 2—figure supplement 7 (number of enriched sequence motifs from mRNA targets that match the paired sRNA) also belongs in the main figures. I suggest combining this with a couple of the most interesting examples of newly found motifs (i.e. unique to this study).

Figure 2—figure supplement 7. The criteria used to make the yes/no calls should be described in the legend.

Figure 3—figure supplement 3. This could be moved to the main figures. The legend needs to be expanded for panel C.

Figure 4—figure supplement 4. I would not expect to see sufficient overlap in regulation between *E. coli* and *Salmonella* for this analysis to be informative. I suggest removing this figure.

Figure 4—figure supplement 8. Panel labels are wrong in the legend.

---

## [Author Response]

While the reviewers are enthusiastic about the potential of the resource that the CLASH data represent, concerns were raised about the validation of these data. Additionally, the reviewers felt that the follow-up studies are interesting, but that some of the conclusions need to be softened. With additional validation of the CLASH data, the manuscript would likely be suitable for publication in eLife, without the need for much in the way of new experimental data. Nonetheless, the required analyses will likely take some time. We encourage you to resubmit if you can make a more compelling case that the CLASH data represent physiological RNA-RNA interactions.

We were extremely pleased with the opportunity to submit a revised version of the manuscript. We have performed a large number of additional bioinformatics analyses to demonstrate the robustness of the CLASH data, focussing also on chimeras that uniquely identified in our CLASH data and those that are supported by a relatively low number of reads.

The requested analyses took longer than expected as we discovered an annoying bug in our bioinformatics pipeline that resulted in some of the reads being incorrectly assigned as chimeras. In brief, we used a software package called FLASH (https://github.com/dstreett/FLASH2) that merges overlapping paired-reads together into one single contig, which are then sent to the hyb pipeline to detect chimeric reads. Those paired reads that FLASH did not consider to be overlapping were subsequently concatenated by our pipeline to make sure that we would also be able to recover chimeras from non-overlapping reads. However, we recently discovered that FLASH is less sensitive in detecting overlapping reads than we expected. As a result, the pipeline concatenated many overlapping paired reads, which hyb then subsequently identified as chimeras. As a result, 10% of the chimeras called by hyb were false positives as they contained almost identical sequences that were frequently annotated as intramolecular interactions. Therefore, we removed these false-positive chimeras and reanalysed the data. This did not change the interpretation of the data. On the contrary, it improved the results substantially. We have included these additional filtering steps in the Materials and methods section of the revised manuscript.

The major concern is that there is currently insufficient evidence to conclude that the RNA-RNA pairs identified by CLASH represent bona fide RNA-RNA pairs inside cells. Many of the reported RNA-RNA pairs appear to have been identified only once, and many include highly abundant RNAs (i.e. tRNA, rRNA). Moreover, overlap with RIL-seq data is fairly limited. While the discussion clearly lays out why overlap with RIL-seq data might be low, this also raises the bar for validating the RNA-RNA pairs not found by RIL-seq. It should be possible to use bioinformatic analyses to further test whether the novel RNA-RNA pairs are genuine. For example, are novel sRNA targets enriched for sequences complementary to the sRNA seeds? Are known sRNA-regulated genes enriched for sRNA-mRNA pairs, and vice versa? These analyses are most important for the novel RNA-RNA pairs identified by CLASH (i.e. not found by RIL-seq).

To address these points, we did a number of additional bioinformatics analyses. These analyses as well as the new results are described in the Results section and presented in Figure 2—figure supplements 1 to 5. In the first supplementary Figure we show that for almost all the sRNA and mRNAs identified in our data that have experimentally verified seed regions we indeed recovered the known seed sequences in the chimera fragments. Moreover, we show that the data has low background levels, as judged by the percentage of rRNA chimeras in the dataset. We obtained very similar results when we repeated these analyses for all the chimeras (Figure 2—figure supplement 2) and the chimeras uniquely identified in the CLASH data (Figure 2—figure supplement 3). The group of chimeras supported by a low number of reads (<4) also largely consisted of sRNA-mRNA fragments as well as sRNA-sRNA and mRNA-mRNA fragments. Moreover, for the vast majority of sRNA chimeras with low read counts we *again* recovered the known sRNA seed sequences. The number of chimeras containing rRNA fragments is slightly higher in this group (12-13%), suggesting higher background. However, considering the sheer abundance of rRNAs in cells (up to 80%) and the fact that we do not do any rRNA depletion step before library preparation, we would argue that the background is remarkably low. Finally, to test whether the novel sRNA targets are enriched for complementary sequences, we folded the sRNA-mRNA chimeras in silico using RNADuplex from the Vienna package and compared it to chimeric reads in which the fragments were randomly shuffled over the same gene or the same class of genes (i.e. all sRNAs or mRNAs). These analyses revealed that the vast majority of chimeras, even those supported by only a few reads (Figure 2—figure supplement 5B-D), had a significantly higher propensity to form stable duplexes compared to randomly generated chimeric reads (p-value < 6*10^-16^).

Collectively, these data strongly suggest that the vast majority of chimeras that we recovered, including the new interactions and the less abundant interactions, represent genuine base-pairing interactions rather than random ligations.

Reviewer #1:Overall, this is very interesting work, and the manuscript obviously represents a great deal of effort. My major criticisms of the work are two-fold. First, the manuscript seems very diffuse – touching on too many topics at a rather surface level. Second, the biological implications of the experimental results are overstated. I hope my comments will be useful to the authors as they consider how to revise their manuscript.1) The title of the manuscript is misleading. The main functional characterization of MdoR and its targets is intriguing and hints at a physiological function related to carbon source adaptation, but there is a long way to go to say that this is truly the function of this sRNA.

We have now changed the title to “Hfq CLASH uncovers sRNA-target interaction networks linked to nutrient availability adaptation**”**. We hope that this title now better reflects the experimental data.

2) ArcZ-CyaR experiment in the middle is not well connected to the rest of the manuscript. The inclusion in the model figure doesn't really help shed light on the biological role for this interaction.

We agree that this experiment does not blend in well with the rest of the manuscript, but we felt we had to do some validation of the identified sRNA-sRNA interactions to demonstrate that the sRNA-sRNA interactions that we have identified are biologically relevant. We have moved the results describing the validation of the ArcZ-CyaR interaction to the Supplementary data.

3) Subsection “Hfq CLASH predicts sRNA-sRNA interactions as a widespread layer of post transcriptional regulation”, third paragraph. Figure 5D, the wild-type ArcZ still affects mutant CyaR levels. The authors provide a hand-waving explanation that could be tested.

We agree that it is strange that the wild-type ArcZ can still affect mutant CyaR levels, but we believe this is because we did not sufficiently disrupt the base-pairing interaction potential. However, the observation that the compensatory mutations in CyaR restore the regulatory activity of the ArcZ seed mutant does support the idea that the two sRNA physically interact in vivo. As stated above, we have now moved these data to the Supplementary data.

Moreover, the authors state that ArcZ promotes CyaR degradation, but there is no direct evidence for this. It could be tested. Not sure it's the highest priority for this manuscript, given that this experiment in general is not well integrated. But at least the authors should modulate their statement to reflect the actual data.

We have removed the text where we state that ArcZ promotes CyaR degradation. We now statein the Results section: “These results, together with the CLASH data, imply that ArcZ and CyaR base-pair in vivo, and that this interaction could lead to a reduction in CyaR levels but not vice versa.” .

4) Abstract – there is no direct evidence that MdoR enhances maltose uptake.

We now state in the Abstract that: “We hypothesize that MdoR contributes to the rearrangements in the outer membrane necessary for efficient uptake of maltose/maltodextrins”.

5) I did not understand the logic behind the analyses in Figure 2. The authors state that it was "logical to assume that changes in Hfq binding would also be reflected in changes in sRNA steady state levels." However, there are numerous studies showing that different sRNAs bind Hfq via different modes, and that there is a great deal of variability regarding the role of Hfq in stabilizing sRNAs. Moreover, the competition among RNAs for binding to a limiting pool of Hfq will certainly change over time, and be influenced by the total sRNA abundance and any given sRNA's proportion of the total RNA pool. There seems to be no overall conclusion from the figure, and no follow up, so I would recommend deleting it.

As requested by the reviewer, we have removed these analyses from the manuscript.

6) Subsection “MdoR directly regulates the expression of major outer membrane porins and represses the envelope stress response pathway”, fourth paragraph: The authors state hypotheses in this section that are not further tested, and are not supported by data shown. These are more appropriate for modest speculation in the Discussion.

As requested by the reviewer, we have moved this to the Discussion section.

7) Figure 7F: Is the effect of MdoR SM on MicA significant?

In this dataset the effect on MdoR SM on MicA was not statistically significant (p-value = 0.06). However, to generate more convincing results, we repeated the qPCRs and included an additional biological replicate experiment. We also repeated the other qPCRs as the *rpoE* and RyeA data were noisier than the other samples. We have now also added p-values to all the bar plots. We feel that the new results more convincingly show that MdoR suppression of MicA relies on the MdoR seed sequence. We do note that the MdoR seed mutation still results in down-regulation of RyeA, suggesting that the regulation is indirect or relies on other sequences within MdoR. We now discuss this in the subsection **“**MdoR directly regulates the expression of major outer membrane porins and represses the envelope stress response pathway”.

8) The only MdoR-target interaction that was definitively demonstrated was MdoR-ompC, and indeed, the authors went above and beyond with evidence here. It is interesting that ompC levels are reduced in maltose (Figure 8B), but this is clearly NOT MdoR-dependent (Figure 8D). The differences in MicA and lamB RNA levels in the mdoR mutant grown in maltose are intriguing, but these effects can't be linked to a specific MdoR-target regulation. Minimally, the authors should try to make the link between molecular interaction of MdoR and a target (rpoE?) and the differences in MicA/lamB more clear.

Our initial hypothesis was that MdoR would help to suppress MicA levels by directly targeting *rpoE*, but we did not find any evidence for direct interactions in our data or the RIL-seq data. However, to make the link between MdoR, *rpoE* and MicA clearer, we repeated the qPCRs of the data shown in Figure 6F and included a third biological replicate so that we could get more reliable statistics. These data show that, consistent with the DESeq analyses, *rpoE* levels do go down upon over-expression of MdoR, however, the changes in *rpoE* levels are not statistically significant. Secondly, we also performed qPCR on *rpoE* levels in RNA samples extracted from the strain that has seed mutations in chromosomal copy of MdoR. We assumed that if the increase of MicA in the MdoR seed mutant strain was directly linked to *rpoE*, we would also see also higher *rpoE* mRNA levels in this strain. This was not the case (see revised Figure 7E). Therefore, our current model is that MdoR enhances LamB expression by suppressing MicA independently of *rpoE*. This then begs the question whether MdoR *directly* targets MicA. Unfortunately, MdoR-MicA chimeras were not found in our Hfq CLASH data and the base-pairing interactions predicted by RNA co-fold and RNADuplex from the Vienna package were not at all convincing, so we have not yet been able to address this question. We now discuss these new findings in the Discussion section.

9) Subsection “MdoR enhances maltoporin expression during maltose fermentation”, last paragraph: It would be very exciting if the data directly supported this statement. However, the experiments presented fall short. More physiological evidence is needed – growth phenotypes, maltose uptake assays, etc. In the absence of these, the authors must tone down their claims.

We have removed this sentence from the manuscript and now present a hypothesis about the role of MdoR in nutrient adaptation: “Based on these results, we hypothesise that when cells decide to use maltose as a main carbon source, subsequent MdoR expression enhances the uptake of maltose by suppressing MicA expression, independently of *rpoE*. This in turn enhances the production of the LamB maltoporin (Figure 8A)”. We believe that, based on the data, this is a reasonable hypothesis. With respect to physiological evidence, we now acknowledge in the Discussion section that this is indeed lacking and present an idea on how to pursue this.

10) Because there is so little investigation of the physiology, the discussion of the physiological relevance of these findings is very superficial. The transition from exponential to stationary phase growth has been studied in *E. coli* growing in LB. What becomes limiting? The authors say very generically "the most favorable nutrients" become limiting. The finding that malEFG and MdoR are specifically expressed during a very narrow window of time in LB grown cells is very interesting. There must be more to the story of their regulation than malT-dependent maltose-inducible expression given this expression pattern in LB given that the main carbon source in LB is peptides/amino acids. The authors should work on improving the quality of the discussion of these issues, and be up front about the limitations of their study in this regard.

We acknowledge that we have not presented evidence demonstrating the physiological relevance of MdoRWe agree that the transient MdoR expression in LB may not be exclusively dependent on the availability of maltodextrins – given the complexity of *E. coli* physiology in LB (e.g. pH, cell density, concurrent metabilization of other substrates etc) and that this is worthwhile investigating. We initially prioritised the study of nutrient-dependent aspect of MdoR function because MdoR accumulates after glucose depletion and the few remaining carbohydrates in LB are utilised sequentially – and during this process MalT is induced. We now discuss in detail which nutrients become limiting in the Discussion section. Moreover, the MdoR expression pattern is similar to that of the other MalT-regulated genes – thus MalT control is key for understanding MdoR physiology. Thus, we considered it important to dissect separately the role of MdoR when maltose is the sole carbon source. It is also known that induction of the *mal* operon is not as effective in a medium that contains other carbon sources (Zhou et al. BMC Systems Biology 2013).

11) One key issue that should be addressed in the Discussion is the fact that these global approaches have so little overlap. I did appreciate the thorough description of the relative advantages provided by the Hfq-CLASH method as compared to RIL-seq. However, I think the field as a whole needs to find a way to discern direct, physiologically-relevant interactions from those that may be transient, weaker, and stochastic. I don't expect the authors to solve this issue, but it should be acknowledged. The sensitivity and accuracy of various methods needs a thorough investigation.

We completely agree that a thorough comparison between the various methods should be done to test the sensitivity and accuracy of each method. In our opinion, this should ideally be done with multiple identical samples that are then analysed in several labs that are using these methods. We would be very keen to contribute to such a study so that the field as a whole can come to a consensus on best practices for performing these type of studies.

It is not possible to quantify the number of interactions that are the result of transient/stochastic/weak interactions, however, we acknowledge that they will certainly be present in our data. We do predict that the frequency of such interactions in our data will be low. This is now discussed in detail in the Discussion section, but we would like to point out here that the CLASH data is highly enriched for chimeras that form stable duplexes, even those chimeras supported by only a few reads (Figure 2—figure supplement 5). Therefore, we would predict that the majority of interactions that we recover represent genuine base-pairing interactions but to what extent these are functional is of course not possible to deduce from the data. We now also discuss ways that would enable us to systematically test which sRNA-mRNA interactions could be biologically relevant.

At least, the authors could consider their Hfq-CLASH results in light of their total expression profiles (RNA-seq) of well characterized sRNAs and their regulons.

This is an excellent point and we had looked at the correlation of CLASH hits and target expression in detail (see Author response image 1). For these analyses we specifically focussed on known interactions with GcvB and CyaR (Author response image 1) as well as novel interactions with these sRNAs (Author response image 1). Although in some individual cases we could see a positive or negative correlation between steady state levels (RNA-seq; TPM values) and number of chimeras, the overall results were unsatisfying as we could not detect a clear pattern. This may have to do with the fact that for a number of mRNA interactions we did not identify many chimeras, which would make the comparison with RNA-seq steady state levels problematic. We decided therefore not to include these analyses in the manuscript.

What's the false negative rate for known interactions?In response to the comments, we have done a significant amount of additional bioinformatics analyses to evaluate in more detail the quality of our data (see Figure 2—figure supplements 1-5). Here, we also focussed on experimentally verified sRNA-mRNA interactions identified in our data (Figure 2—figure supplement 1). These results show that in all but one case (Gcvb-*sstT* interaction), we recover the known sRNA and mRNA seeds (as well as some potentially new seed sequences). Therefore, the false-negative rate, which we define as the number of incorrect wrong seed sequences identified for known interactions in our data, seems to be low. We now discuss these data in the Results section.

Reviewer #2:This manuscript analyses the RNAs associated with the RNA chaperone Hfq in Escherchia coli at different growth stages, and in particular during the transition between stages. There has been other work published in this topic, but the new aspect of the work presented here is the depth of analysis of the transitions and the in depth characterisation of the associated RNAs. One important finding from this study is that sRNA expression does not correlate strongly with Hfq binding profile – suggesting that there must be context dependent binding of the RNA to Hfq. Another is the model for the regulatory network involving the processed transcript from the mal operon. The experimental work is extensive and there are many interesting new findings reported. There are several comments listed below that will hopefully be useful for the authors to consider:1) Hfq for CLASH has two large tags on C-terminus. As the C-terminus has been proposed to participate in RNA/protein partners binding and Hfq autoinhibition (work from the Woodson group), have the authors done any controls to make sure this does not interfere with RNA banding/introduce false results?

This is an important point and we addressed this in a previous paper (Tree et al. Molecular Cell; see Figure 1—figure supplement 1 and first paragraph of the Results section). The data indicated that HTF tagged Hfq is functional and facilitates MicF repression of OmpF.

2) "Hfq binds to sRNA-target RNA duplexes" – are RNA duplexes the only Hfq targets? For example, sRNAs were shown to cycle on Hfq, therefore one can imagine a situation in which one sRNA is not fully displaced and the second one already bound. Could some of the sRNA hybrids represent such state?

It is certainly possible that such chimeras can be formed during the ligation step. Although it is not possible to quantify the number of such interactions, based on the following we would argue that these interactions are probably not very abundant in our data. Firstly, we purify Hfq and cross-linked RNAs under very stringent and completely denaturing conditions before we do the intermolecular ligation reactions. We would predict that most of such interactions, including weak and stochastic interactions, would dissociate under these conditions. Furthermore, because our purification conditions completely disrupt the Hfq hexamer (this work and (Tree et al., 2014)), such transient interactions would only be detected if an Hfq *monomer* was UV cross-linked to both sRNAs simultaneously and if the available 5’ end 3’ ends are in close proximity. Considering the low efficiency of UV cross-linking, the likelihood of this happening is very low. Secondly, we show that our chimeras, including those chimeras that are supported by only a few reads, are highly enriched for stable duplexes as well as sRNA seed sequences. It seems therefore unlikely that the CLASH protocol would efficiently recover weak and or transient interactions. We now discuss this in the Discussion section.

3) 'tRNA-tRNA and rRNA-rRNA chimeras originating from different coding regions were removed' why?

There are many copies of tRNA and rRNA genes in the genome that are very similar, and the sequence aligner cannot always determine from which copy the two halves of a chimera came from. So, in many cases our software will label these rRNA-rRNA and tRNA-tRNA chimeras as “intermolecular” because each half was mapped to a different copy. But we cannot rule out the possibility that these fragments originated from the same gene. Therefore, we consider these to be false positives as they are very likely intramolecular interactions. We now mention this in the legend of Figure 2A.

4) Can the authors please comment on other chimeras isolated, others than sRNA-mRNA and mRNA-mRNA? Would these represent Hfq targets in the cell?

A few % of the chimeras appear to represent sRNA-tRNA interactions. Although it is unclear whether these are biologically relevant, it is worth noting here that in *E. coli* external transcribed spacers of tRNAs can base-pair with sRNAs to absorb transcriptional noise (Lalaouna et al., 2015). Moreover, the predicted base-pairing interactions between the tRNA and sRNA halves in chimeras are in many cases quite extensive (Supplementary file 2). Between 4-5% of the chimeras represented sRNA-rRNA interactions. Binding of Hfq to rRNA has also been demonstrated (Andrade et al., 2018), however, this interaction appears ot be independent of sRNAs. Therefore we predict that sRNA-rRNA interactions likely represent noise. However, considering the sheer abundance of ribosomal RNA in a cell (80%) we would argue the noise is quite low.

We now discuss this in more detail in the Results section and we have included more extensive analyses of all the types of interactions in Figure 2A and Figure 2—figures supplements 1-5.

5) Figure 3 – it is not clear what the enriched motifs are showing, 5' end of the chimera? 5' end of both RNAs in the chimera? Only mRNAs?

We apologise for the confusion. Only the mRNA fragments in chimeras were considered for the motif analyses. The left panel shows motifs of the 5’UTRs of mRNAs found in chimeras, the right panel, the motifs of the 3’UTRs of mRNAs found in chimeras. We have now added a sentence to the figure as well as the figure legend to make this clearer.

6) Explain in more detail what is meant by scrambled RNA.

This an sRNA that has a random nucleotide sequence. We have now made this clearer in the figure legend. The control plasmid is pJV300, the standard control plasmid for pL-driven sRNA expression. It expresses a ~50 nucleotides long nonsense RNA derived from *rrnB* terminator region of the backbone plasmid.

7) Figure 4 – as only mutations in ArcZ cause disruption of the regulation, can it be an indirect effect, not the result of direct sRNA-sRNA regulation?

We cannot rule out that the regulation is indirect, however, the fact that the compensatory mutations in CyaR restore the regulation of the ArcZ mutant is strong evidence that the regulation is direct. Note that we have now moved these analyses to the supplementary information, as requested by reviewer #1

8) Figure 5B – It is difficult to see expression of ygaM (or YgaN, which the blot may be showing).

(Note Figure 5 is now Figure 4) We apologize for the fact that YgaN is poorly detectable. It is clearly a very low abundant fragment and even after exposing the blot for two weeks with several probes, we were not able to get a stronger signal. This is why we added a larger scan of the same blot in Figure 4—figure-supplement 1A, where ygaN is more visible.

Where is MdoR on the blot? If it is labelled malG it is somewhat confusing, as the blot presumably shows the sRNA fragments, not the whole mRNAs?

We apologise for the confusion. What is shown in Figure 5B (now Figure 4B) are indeed the sRNAs, but we had labelled those sRNAs derived from 3’UTRs with the names of the host mRNA. We now included the sRNA names.

9) The signal for RyhB on Figure 5B is quite strong for OD 1.2 and 1.8, however on Figure 6C it is very weak. Can the authors explain? MdoR intensities seem to match, so presumably the RNAs quantities used are similar?

We had to expose the blot for about 4-5 days to get a good MdoR signal and by the time we hybridized the blot with the RybB probe the ^32^P label was already about two weeks old. This explains why the signal is a bit weaker. CpxQ and 5S rRNA probing was performed with fresh label once the signals on the blot had sufficiently decayed.

10) Is there an evidence that RyhB is in the cell as 5'PPP RNA? Perhaps it is not processed, but has the possibility of it harbouring a different 5' end has been excluded?

RybB was not reported to be processed by RNase E for maturation (Chao et al., 2017) and it does not have a 5’NAD modification (Cahová et al., 2015). RybB is also TEX insensitive (Thomason et al., 2015). In this study, the 5’PPP ends were converted using TAP to 5’P after the TEX step. RybB was also detected in the TEX untreated sample, so we can infer it originally harboured a 5’PPP.

11) Figure 7C – It is confusing that the authors label 5' and 3' ends which are not real ends, and are different for each panel for MdoR. Could they mark, e.g. with dots, that these are not real ends of RNAs? Or indicate positions of the nucleotides shown? It would make analysing the results much easier.

As suggested by the reviewer we have now marked the ends with dashed lines to indicate that these are not the beginning and ends of the RNAs. Note that the original Figure 7 is now Figure 6.

12) Figure 7D- If an empty plasmid is used as a control and the blot probed for MdoR, can the authors explain what is being expressed in their control after 20 minutes? There may be a typo in the legend as it states that the samples were harvested 15 minutes after induction, but the blot shows the results for 20. What is the meaning of the red rectangle over the 15 minutes into MdoR expression?

The RNA expressed in the control after induction is the endogenous MdoR (15 min after induction, cells are near OD_600_ 0.8, the OD_600_ at which MdoR starts being expressed). The plasmid that we used to over-express MdoR has two transcriptional terminators, one from the operon itself and one already present in the plasmid. We believe that the longer band represents MdoR terminated at the second terminator. The red rectangle indicates the time-point for which RNAseq and differential expression analyses were performed. Made this clearer in the main text and the figure legend.

13) Figure 7F – explain MdoR SM, it also isn't introduced in the text. Why does the seed mutation cause higher target levels? For RyeA it doesn't seem like the seed mutation has abolished regulation. Is RybB regulating MicA as well?

MdoR SM indicates the MdoR seed mutant. We apologise for not including this in the text. This has now been fixed. As stated above (# reviewer 1 point 7), we have repeated the qPCRs presented in this figure and added an additional biological replicates to make the data more robust. The new data confirm that the MdoR seed mutant does not abolish RyeA regulation. We now discuss this in the subsection “MdoR directly regulates the expression of major outer membrane porins and represses the envelope stress response pathway”. We do not believe RybB directly regulates MicA (or *rseA*; Figure 6F) but that the decrease in their RNA is the result of a homeostatic negative feedback loop, as proposed by the Gottesman lab (Thompson et al., Journal of Bacteriology, 2007; https://jb.asm.org/content/189/11/4243.long). A recent paper from Sarah Ades lab (Nicoloff et al. Journal of Bacteriology 2017) suggests that the RybB levels need to be carefully controlled as too high expression can be toxic under some circumstances. One way to respond to RybB over-expression would be to reduce transcription of the sigmaE operon, which includes *rpoE* and RseA.

14) Have the authors tested how their substantial MdoR seed mutation influences RNA structure? Is it possible that, as the seed seems internal, the overall structure of the sRNA is disrupted and therefore the regulation lost? Can the mutant still bind to Hfq? The structure change would also explain problems with RNase E processing.

This is an interesting point and we have not tested it. Our initial idea was to make a knock-out of MdoR, but we were worried that this would also impair MalG expression. Therefore, we decided to make a mutant version that we predicted would completely disrupt the base-pairing interaction of MdoR with its targets but would not affect accumulation of MalG. The fact that it is no longer cleaved by RNase E was actually a bonus is because we wanted to remove the sRNA entirely. We therefore did not test whether the transcript was still bound by Hfq or its secondary structure.

15) 'Notably, the fully-processed mutant MdoR sRNA is less abundant than the wild-type (Figure 9C) and longer (unprocessed) fragments that contain upstream malG regions could be readily detected (Figure 9E) '- should be 8C and 8E

We thank the reviewer for pointing this out. This has been corrected. Note that the data is now presented in Figure 7.

16) 'We conclude that the dynamics of sRNA expression and binding to Hfq are not always highly correlated.' Any thoughts why?

We hypothesized that this may be linked to the availability of Hfq, as the protein is ~15 times less abundant at exponential phase as compared to stationary. Similar changes in Hfq expression at different growth phases have also been observed in pathogenic bacteria. It is conceivable that at different growth stages sRNAs are packaged in different RNPs and that the composition of these complexes is dynamic. The *E. coli* sRNA IsrA/McaS has been shown to associate with a large number of different proteins, including Hfq, ProQ and CsrA; RNA chaperones that are known to bind and stabilize sRNAs and regulate sRNA-target interactions. It is tempting to speculate that the composition of this RNP may be growth phase dependent. Therefore, Hfq may not be essential to stabilize all sRNAs at low cell densities and their stability may vary at different growth stages. A plausible model is that some sRNAs are sequestered and sufficiently stabilized by other RBPs (such as ProQ and CsrA) during early growth stages and that Hfq can only stably associate with these RNPs once expression levels are sufficiently high. However, reviewer #1 recommended that we remove these results from the manuscript and therefore these data are no longer included.

17) Polysome preps used cyclohexamide, but this acts but blocking the peptide exit channel in the ribosome and may not trap polysomes except by blocking the last ribosome on the assembly. Another antibiotic or non-hydrolysable GTP might be better.

We trapped ribosomes on mRNAs using two combined approaches. One is to add cycloheximide, the other was to flash freeze *E. coli* samples and pulverize them under liquid nitrogen. The latter procedure is known to be most conservative of polysomes without introducing biases or artefacts. Indeed, the use of the eukaryotic elongation inhibitor cycloheximide (CHX) is under debate for ribosome profiling (not for polysomal profiling), as it may introduce artefacts in ribosome positioning along the transcripts (Gerashchenko and Gladyshev, 2014). In our manuscript we performed polysomal profiling and not ribosome profiling. Moreover, we analysed the data as differential uploading of transcripts between two conditions, making it is reasonable to assume that any possible bias induced by the drug are negligible.

18) These references have related information that may be useful to comment on in the manuscript: de Mets, van Melderen and Gottesman, 2018; Miyakoshi et al.,.

We now discuss the 3’UTR-derived SdhX sRNA in the subsection “

Hfq CLASH identifies novel sRNAs in untranslated regions” where we describe chimeras supported by a low number of reads. We found 2-3 chimeras with SdhX and known interactions (*katG* and *ackA*) and we show that the predicted secondary structure of the chimeric reads matches the known interaction reported in these papers.

Also, Hfq has been known to be involved in nutrient uptake regulation in Pseudomonas aeruginosa, where it inhibits translation of certain mRNAs depending on which nutrients are available. Pei et al., 2019, have solved high resolution structures of Hfq in complex with a target mRNA and other effector molecules to show how this Hfq based regulatory complex works. This research may be related to the theme of the report here and it might be helpful to comment on these findings.

We agree that this is a very interesting of Hfq-dependent regulation of gene expression and relevant to our work. We now mention in the Introduction that Hfq can also regulate gene expression independently of sRNAs and we mention the Hfq-Crc example in *Pseudomonas*.

[Editors’ note: further revisions were suggested prior to acceptance, as described below.]

All the reviewers were enthusiastic about the manuscript and recommend acceptance pending some edits to the text. In particular, the reviewers felt that the new analyses of the CLASH data make a strong case that the identified RNA-RNA interactions are real, and thus greatly expand the known set of interactions for *E. coli*, and reveal important insights such as the abundance of sRNA-sRNA interactions. To better focus the manuscript, we recommend removing the section on MdoR. While the reviewers found this work to be of interest, they also felt that it was peripheral to the main theme of the study, and would be better suited to an independent publication in a more specialized journal. This would free up some space in the paper to move some of the supplementary figure panels into the main figures, improving readability. Reviewer 3 has some specific suggestions for supplementary figure panels that could be moved into the main set of figures. The detailed reviews are listed below:Reviewer #1:The authors have provided further experimental data and analysis and made compelling response to most of the points raised in the review. The manuscript has been improved and the support for the conclusions strengthened considerably.One minor issue is the Figure 2E legend does not explain the figure very clearly.

We apologise for not explaining this properly. We have improved the explanation in the figure legend.

Reviewer #3:The new analyses of the CLASH data make a very convincing case that the novel RNA-RNA pairs reflect real in vivo interactions. My preference would be to remove the MdoR story, which is interesting but peripheral to the main theme of the paper, and does not look at a novel sRNA (MdoR was identified previously by RIL-seq).

We have now removed all the data referring to MdoR and will include this in a different manuscript. Because of this, we had to make a number of changes to the text, including rewriting the Abstract and the last paragraphs of the Introduction. In these sections we now focus more on our findings that interactions that are more reproducibly recovered are more likely to have a regulatory outcome and that base-pairing potential is important but it does not have the strongest predictive power. Note that we have now renamed MdoR MalH in the main text and all of the figures as this is more in line with the nomenclature in the field.

Moreover, I suggest moving some of the more important supplementary figure panels into the main part of the paper.Figure 2—figure supplement 1C. The "distance from sRNA seed" numbers appear to be similar to the length of the sRNAs. The authors should indicate the sRNA lengths.

We have added the lengths of the sRNAs to the heat maps shown in the supplementary figures associated with Figure 2.

Figure 2—figure supplement 6 (predicted base-pairing strength for identified interactions). This is an important analysis and should be moved to the main figures.

We have now included this in the main figures as Figure 3.

Figure 2—figure supplement 7 (number of enriched sequence motifs from mRNA targets that match the paired sRNA) also belongs in the main figures. I suggest combining this with a couple of the most interesting examples of newly found motifs (i.e. unique to this study).

This figure, as well as two examples of complementary sequence motifs identified specifically in our study is now included in the main figures as Figure 4.

Figure 2—figure supplement 7. The criteria used to make the yes/no calls should be described in the legend.

We have now added this to the legend of Figure 4 where these data are described. An sRNA was considered to have an enriched motif if a motif identified by MEME had an E-value <= 0.1 and/or the MAST p-value of the motif, which indicates the overall match between the identified motifs and the sRNA sequence, was <= 0.001.

Figure 3—figure supplement 3. This could be moved to the main figures. The legend needs to be expanded for panel C.

This is now Figure 7 in the main figures.

Figure 4—figure supplement 4. I would not expect to see sufficient overlap in regulation between *E. coli* and Salmonella for this analysis to be informative. I suggest removing this figure.

We have removed the figure from the manuscript.

Figure 4—figure supplement 8. Panel labels are wrong in the legend.

We have corrected the legend.